



**Intensified Aleutian Low induces weak Pacific Decadal Variability**
William J. Dow[1], Christine M. McKenna[1], Manoj M. Joshi[2], Adam T. Blaker[3], Richard Rigby[1],
Amanda C. Maycock[1]
[1]School of Earth and Environment, University of Leeds, Leeds, UK
[2]Climatic Research Unit, School of Environmental Sciences, University of East Anglia, Norwich,
UK
[3]National Oceanography Centre, Southampton, UK
**Abstract**
The Aleutian Low drives decadal variability in North Pacific sea surface temperatures (SST), but
its role in basin-wide Pacific SST variability is less clear owing to the difficulty of disentangling
coupled atmosphere-ocean processes. We apply local atmospheric nudging to isolate the effects
of an intense winter Aleutian Low using an intermediate complexity climate model. An intensified
Aleutian Low produces a basin-wide SST response with a similar pattern to internally-generated
Pacific Decadal Oscillation (PDO). The amplitude of the SST response in the North Pacific is
comparable to PDO, but in the tropics and southern subtropics the anomalies induced by the
intense Aleutian Low are a factor of 3 weaker. The tropical Pacific warming peaks in boreal spring,
though anomalies persist year-round. A heat budget analysis shows the northern subtropical
Pacific SST response is predominantly driven by anomalous surface heat fluxes in boreal winter,
while in the equatorial Pacific the response is mainly due to meridional heat advection in boreal
spring. The propagation of anomalies from the extratropics to the tropics can be explained by the
seasonal footprinting mechanism, involving the wind-evaporation-SST feedback. The results
show that low frequency variability and trends in the Aleutian Low could contribute to basin-wide
anomalous Pacific SST, but the magnitude of the effect cannot explain the full amplitude of the
PDO. This finding suggests that external forcing of the Aleutian Low is unlikely to explain observed
shifts in the phase of PDO in the late 20th and early-21st centuries.




Key points (140 chars)

1. Relaxing towards a strong winter Aleutian Low produces warming across the equatorial
Pacific that peaks in boreal spring.
2. Changes to surface heat fluxes (subtropics) during boreal winter and meridional advection
(equatorial) during boreal spring in the upper ocean drive the SST warming.
3. A combination of the seasonal footprint mechanism and wind-evaporation-SST
mechanism generate the surface climate anomalies in the tropical Pacific.







## 1. Introduction

The Aleutian Low has a well-known role in determining the North Pacific component of the Pacific Decadal Oscillation (PDO) (e.g. Schneider and Cornuelle, 2005; Zhang et al., 2018; Hu and Guan, 2018; Sun and Wang, 2006; Newman et al. 2016). Fluctuations in the Aleutian Low intensity affect the North Pacific subpolar gyre (Pickart et al. 2008), upper ocean temperatures (e.g. Latif and Barnett, 1996) and sea surface height (Nagano and Wakita, 2019) through anomalous thermal forcing and wind stress. Oceanic Rossby waves initiated by Aleutian Low variability can propagate westward and cause lagged signals in the Kuroshio-Oshashio Extension (KOE) region (e.g., Kwon and Deser, 2007).

The prevailing paradigm for the PDO regards the role of the Aleutian Low to be largely driven by tropical processes via excitation of upper tropospheric Rossby waves (Newman et al. 2016; Zhao et al. 2021; Vimont. 2005; Knutson and Manabe 1998; Jin 2001). However, decadal changes in the Aleutian Low may arise via other mechanisms including Arctic sea ice trends (Simon et al. 2021; Deser et al. 2016), Arctic stratospheric variability (Richter et al., 2015), or as a local response to external forcings (Smith et al. 2016; Dow et al. 2021; Dittus et al. 2021; Klavans et al. submitted). It has been proposed that observed shifts in the PDO in the late 20th and early 21st centuries were driven by anthropogenic forcing of the Aleutian Low, which was then communicated to a basin-wide PDO signal (Smith et al. 2016; Klavans et al. submitted). However, the mechanisms via which North Pacific anomalies linked to decadal Aleutian Low changes may be communicated into a basin-wide SST response, and whether the amplitude of such a response matches observed variations, remain unclear.

Several studies have investigated the North Pacific influence on the tropics using surface flux restoring in a model (Alexander et al. 2010; Sun and Okumura 2019; Liu et al. 2021). Alexander et al. (2010) and Sun and Okumura (2019) imposed surface flux anomalies derived from the North Pacific Oscillation (NPO) - the anomalous North Pacific pattern projecting onto the second EOF of low frequency tropical Pacific SST variability. They showed that surface forcing associated with the NPO can affect decadal variability in the tropics. The proposed mechanism for communication of extratropical surface anomalies into the tropics is the seasonal footprinting mechanism (SFM) (Alexander et al. 2010; Sun and Okumura 2019; Amaya et al. 2019, Liu et al. 2021). Atmospheric circulation anomalies driven by the subtropical portion of the high latitude SST footprint modulate tropical SSTs through coupled atmosphere-ocean processes, leading to anomalies that persist



through boreal spring-summer. However, the amplitude of the effect on tropical Pacific SSTs from
the North Pacific has been suggested to be quite weak on decadal timescales (Alexander et al.
2010; Sun and Okumura 2019; Liguori and Di Lorenzo 2019). Moreover, the studies did not
directly isolate driving by the Aleutian Low, which has been highlighted in studies arguing a role
for anthropogenic forcing of recent observed PDO variability (Smith et al. 2016; Klavans et al.
submitted).

In this study, we aim to better understand the role of long-term changes in the Aleutian Low in
governing the multi-annual behaviour of tropical Pacific SSTs. We perform an ensemble of
atmospheric nudging simulations in an intermediate complexity coupled climate model to isolate
the effect of an anomalous Aleutian Low and compare this with internally-generated low frequency
Pacific variability in a free running simulation. The manuscript is structured as follows: section 2
describes the methodology and details of the model used. Section 3 compares the results of the
nudging simulations with the free running simulation. Discussion of the results is provided in
section 4 and conclusions in section 5.


**2. Data and Methods**

**2.1 FORTE 2.0**

Simulations were performed using FORTE2.0, an intermediate complexity coupled Atmosphere-
Ocean General Circulation Model (AOGCM). The atmospheric model IGCM4 (Intermediate
General Circulation Model 4) (Joshi et al., 2015) uses a truncated series of spherical harmonics
run at T42 resolution with 20 $\Sigma$-levels to a height of $\Sigma = 0.05$. IGCM4 is coupled to the MOMA
(Modular Ocean Model – Array) (Webb, 1996) ocean model run at 2º x 2º resolution with 15
vertical levels. The two components are coupled once per day using OASIS version 2.3 (Terray
et al., 1999) and PVM version 3.4.6 (Parallel Virtual Machine). As described in Blaker et al. (2021),
between 5º N/S and the equator the horizontal ocean diffusion increases by a factor of 20 to
balance equatorial upwelling and parameterise the eddy heat convergence. For more details on
the model see Blaker et al. (2021). The model simulates low frequency SST variability in the
Pacific with a similar pattern to that seen in observations but a weaker amplitude by around a
factor of 4 to 5 (Figure S1).





### 2.2 Grid-point nudging method

Atmospheric nudging has been used to investigate climate and weather relationships between remote phenomena (e.g. Martin et al., 2021; Knight et al., 2017; Watson et al., 2016). A nudging code was added to IGCM4. Nudging was performed by adding tendencies to horizontal winds, temperature and surface pressure. The nudging code is publicly available at (https://github.com/NOC-MSM/FORTE2.0).

The nudging configuration is similar to that in Watson et al. (2016), with two additional terms to account for vertical (z) and temporal (t) variation in the nudging strength:

$$\delta x(\lambda, \phi, z, t) \ = \ -\gamma(\lambda, \phi, z, t)(x(\lambda, \phi, z, t) \ - \ x_{ref}(\lambda, \phi, z, t))/\tau, \qquad \text{(Eqn 1)}$$

where $x$ is the variable being relaxed as a function of longitude ($\lambda$) and latitude ($\phi$), $x_{ref}$ is the reference state, and $\tau$ is the nudging strength (set to 6hr). The spatial extent of the nudging was tested extensively to avoid any shock at the boundaries and spurious effects of nudging near polar regions. The regional extent was determined as:

$$\gamma(\phi, \lambda) \ = f(\phi, \phi_1, \phi_2)f(\lambda, \lambda_1, \lambda_2), \qquad \text{(Eqn 2)}$$

where

$$f(\phi, \phi_1, \phi_2) \ = \ [1/(1 \ + \ e^{-(\phi-\phi_1)/\delta_1})][1 \ - \ 1/(1 \ + \ e^{-(\phi-\phi_2)/\delta_2})] \ \text{(Eqn 3)}$$

and

$$f(\lambda, \lambda_1, \lambda_2) \ = \ [1/(1 \ + \ e^{-(\lambda-\lambda_1)/\delta_1})][1 \ - \ 1/(1 \ + \ e^{-(\lambda-\lambda_2)/\delta_2})] \ \text{(Eqn 4)}.$$

$\Phi_1$ = 30ºN and $\Phi_2$ = 65ºN represent the southern and northern limits of the nudging region and $\lambda_1$ = 160ºE and $\lambda_2$ = 140ºW are the western and eastern limits of the nudging region. The horizontal limits follow the commonly defined North Pacific Index (NPI) (Trenberth and Hurrell, 1994) as a proxy for the region encompassed by the Aleutian Low.

The strength of the tropospheric nudging is set to 1 at $\Sigma$ = 0.96 (lowest atmospheric level), decreasing exponentially to 0 at $\Sigma$ = 0.05 (tropopause). Nudging is applied during the extended boreal winter season (NDJFM) peaking on 15 January, with a Gaussian function in time to



increase the nudging strength from 0 to 1 between 1 to 30 November and a reverse ramp-down
during March. The spatio-temporal forms of the nudging coefficients are shown in Figure S2.
The strong Aleutian Low state is taken from a 100 year long control run (CONTROL) based on a
winter month with an NPI anomaly of -3.02$\sigma$, where $\sigma$ is the standard deviation calculated over all
winter months in CONTROL. Therefore, the target state represents an extreme intense Aleutian
Low state as simulated in FORTE2.0. $x_{ref}$ comprises the anomaly of this month added to the
daily climatology. A 50 member NUDGED ensemble was generated using initial conditions drawn
from each January 1st of the final 50 years of CONTROL. Each member is integrated for 30 years
with nudging commencing on 1 November of the first year and repeating each winter of the
simulation. Unless otherwise stated, the analysis shows ensemble mean anomalies in the
NUDGED simulation compared to the long-term climatology of CONTROL. Statistical significance
is defined by comparing the responses to the magnitude of internal variability. For CONTROL,
variability is calculated by multiplying the standard deviation of overlapping 15-year means by √2.
The median value of the standard deviation is used and the result is statistically significant at the
95% level if the ensemble mean response lies outside of the bounds ±1.96xSD.

### 2.3 Mixed Layer Heat Budget Analysis


The heat budget of the upper ocean mixed layer (assumed to be 30 m deep) is analysed for the
regions shown by the boxes in Figure 1, where the temperature tendency is given by:
dT/dt = ADV + DIFF$_{vert}$ + DIFF$_{horiz}$ + CONV (Eqn. 5).
Daily tendencies due to advection (ADV), vertical and horizontal diffusion (DIFF$_{vert}$ and DIFF$_{horiz}$)
and convection (CONV) are output from the model. Vertical diffusion represents the contribution
to the mixed layer heat budget from surface turbulent and radiative fluxes. ADV is composed of
zonal, meridional and vertical components:
$$ADV \ = \ u\frac{\delta T}{\delta x} \ + \ v\frac{\delta T}{\delta y} \ + \ w\frac{\delta T}{\delta z} \ \text{(Eqn. 6),}$$
where u, v and w are the zonal, meridional and vertical components of the ocean velocity and
dT/dx represents the local zonal gradient of temperature. We linearize the meridional advection
term to investigate the relative roles of changes to ocean current velocity and temperature
gradient as follows:



$$\left(v\frac{\delta T}{\delta y}\right)' = v'\frac{\delta T_0}{\delta y} + v_0\left(\frac{\delta T}{\delta y}\right)' + v'\left(\frac{\delta T}{\delta y}\right)' \quad \text{(Eqn. 7)}$$
where the subscript 0 denotes CONTROL values and primes denote anomalies in NUDGED.
**2.4 PDO Index**
The PDO index is calculated as the first EOF of monthly SST anomalies, calculated as deviations
from the climatological seasonal cycle, over the region 20-65ºN, 120-260ºE. Before calculating
the leading EOF, the temperature anomalies are weighted by the square-root of the cosine of
latitude to account for the decrease in area towards the pole. The monthly principal component,
corresponding to the PDO index, is normalised by the standard deviation to give it unit variance.
The pattern of temperature anomalies that covaries with the PDO is found by linearly regressing
the time series of the monthly mean temperature anomalies onto the monthly PDO index (Figure
1b).
**3. Results**

*3.1 Surface temperature response*
Figure 1a shows annual mean surface temperature anomalies in NUDGED expressed as a
change per standard deviation (σ) of the PDO index. A horse-shoe pattern of anomalous
temperature extends across the North Pacific, comprising warming in the north and eastern
Pacific and along the west coast of North America and cooling in the western North Pacific/KOE
region. The strongest warming (0.2-0.3 K/σ) is seen over the North Pacific and western North
America. There is weaker (0.02-0.04 K/σ) but statistically significant warming in the eastern and
central equatorial Pacific. The pattern of temperature anomalies in NUDGED closely resembles
unforced multidecadal Pacific variability in CONTROL (Figure 1b). Therefore, a sustained
increase in Aleutian Low strength forces a basin-wide SST response that resembles internally-
generated coupled variability. However, while the extratropical SST anomalies are somewhat
larger in NUDGED, particularly in the subpolar gyre, the tropical Pacific signal is substantially
weaker by a factor of ~3. This indicates that atmospheric forcing by the Aleutian Low alone is not
sufficient to generate a basin-wide SST response that is consistent with the intrinsic variability of
the model. Note the Aleutian Low state in $x_{ref}$ is extreme (-3σ), meaning a more realistic amplitude
for sustained Aleutian Low intensification can be expected to induce a weaker response.



The seasonality of the surface temperature anomalies in NUDGED is shown in Figure 2 separated
for years 1-2, years 3-4 and years 5-30. The initial response to the intensified Aleutian Low is a
warming in the subpolar gyre in boreal autumn (SON). This amplifies in DJF during the peak of
the nudging period, where a tongue of warming extends into the subtropical North Pacific. This
pattern persists into MAM after nudging ceases but is also accompanied by warming in the
eastern tropical Pacific. By JJA, the tropical and subtropical temperature changes have weakened
leaving residual warming in the subpolar gyre that persists into the following winter. The
temperature anomalies over land quickly dissipate due to the low specific heat capacity. A similar
seasonal evolution occurs in years 3-4, but the tropical warm anomaly emerges earlier in DJF
and extends further westward at its peak in MAM. The anomalies in years 5-30 show a similar
spatiotemporal pattern to the first 4 years, suggesting the mechanisms by which the anomalies
manifest do not evolve strongly when the signals are maintained over multi-year timescales. Small
differences between years 1-4 and 5-30 are the extent of the robust signal in the tropical Pacific;
there is a small reduction in the amplitude of the tropical warming in JJA and no significant western
tropical Pacific warming in MAM for years 5-30. The signal of peak tropical warming in MAM in
NUDGED qualitatively agrees with observed low frequency Pacific variability (Figure S1), though
we note that FORTE2.0 shows a narrower band of tropical warming compared to observations.

### 3.2 Mixed layer heat budget

The mixed layer heat budget in the subtropical North Pacific and Niño 3.4 regions shows different
annual cycles in the anomalous temperature tendencies (Figure 3 a,b). The largest anomalous
surface temperature tendency in the subtropical North Pacific occurs during the nudging period
(DJF), whereas the peak warming tendency in the Nino3.4 region occurs in February-April. In the
subtropics in winter, warming from vertical diffusion is offset by meridional advection. In contrast
in the Niño 3.4 region, anomalous meridional advection contributes to a warming tendency year-
round, with the maximum (~0.3 K/month) in MAM. This warming is partly offset by anomalous
vertical diffusion and convection. Meridional advection therefore contributes to cooling in the
subtropical North Pacific but causes warming in the Niño 3.4 region.

The anomalous meridional advection in the subtropical North Pacific is dominated by the change
in meridional velocity, whilst in the Niño3.4 region the change in meridional temperature gradient
is the largest contributor throughout most of the year (apart from Sept-Dec). The enhanced





warming tendency from Feb-June in the Niño3.4 region is driven by changes in meridional
velocity. The difference in contributing terms implies different mechanisms governing the
changing mixed layer temperatures in the two regions.

The net surface heat flux anomalies in NUDGED are shown in Figure 4(a-d). There are positive
net surface heat flux anomalies across the North Pacific and within a SW-NE oriented band in the
subtropical North Pacific. The largest heat flux anomalies occur during DJF, with values in excess
of 4 W m$^{-2}$/$\sigma$. The net surface heat flux anomalies in NUDGED are dominated by the latent heat
flux (Fig. 4 e-h). The pattern of surface latent heat flux anomalies in JJA in the extratropical North
Pacific resembles that for the internal PDO structure (Figure S3), with positive flux anomalies
extending eastward from the KOE region, which are enveloped by negative anomalies in the
northeast Pacific and subtropical North Pacific. The persistence of surface latent flux anomalies
year-round is expected given the surface temperature persistence and alludes to ocean-
atmosphere feedbacks.

*3.3 Atmospheric circulation response*
Figure 5 shows the seasonal mean zonal and meridional near-surface wind anomalies in
NUDGED. As expected, the largest anomalies occur in the period over which nudging is applied
(DJF), with a westerly zonal wind anomaly of up to ~0.5 ms$^{-1}$/$\sigma$ in the subtropics and an easterly
anomaly of a similar magnitude in the subpolar extratropics. The meridional wind shows
alternating southerly-northerly anomalies across the North Pacific orientated with a north-easterly
tilt suggesting a Rossby wave train response. The subtropical zonal wind anomalies project onto
a southerly shift of the westerlies compared to the climatology in CONTROL, with persistent
anomalies extending into the spring after nudging ceases (MAM). Interestingly, there is an
emergence of a westerly wind anomaly near the coast of California in DJF that extends southward
and westward into the equatorial Pacific in MAM. Although zonal wind anomalies are evident in
JJA, they are not strongly statistically significant.
Figure 6 shows the latitude-time evolution of surface temperature, near-surface wind and surface
pressure anomalies in NUDGED averaged over the central and eastern tropical Pacific. There is
year-round warming in subtropical and equatorial regions, with the largest magnitude in the
subtropics from November through April (~0.05 K/$\sigma$) and in the equatorial region from March
through July (~0.3 K/$\sigma$). The nudging invokes concurrent warming in the subtropics, while there





is a seasonal delay in the emergence of warming in the equatorial Pacific. From July to November
in the subtropics (around 15°N) there is substantially less warming than during the rest of the
year, with values close to zero. The westerly wind anomalies coincide with the timing of the
temperature anomalies, with south-westerly anomalies of ~0.05 m s$^{-1}$/$\sigma$ in the subtropics and
~0.03 m s$^{-1}$/$\sigma$ in the equatorial region. In addition to the cross-equatorial temperature gradient
generated by the subtropical anomaly, the lower surface pressure in the northern subtropics (~1.5
hPa), which is largest in February and March, creates a pressure gradient across the equator. At
this time there is evidence of cooling in the southern subtropics (south of 15°S).

**4. Discussion**

The impact of an intensified Aleutian Low on the tropical Pacific in this study suggests an
excitation of the SFM mechanism (e.g. Vimont et al. 2003; Alexander et al. 2010; Chen and Yu,
2020; Sun and Okumura, 2019). In accordance with the SFM, the SST anomalies persist into the
summer season, with anomalous temperatures found in the North Pacific year round. The signals
in winter and spring show a similar spatial signature to that found by Liguori and Di Lorenzo
(2019), who show an SST signature in the subtropics as a precursor to ENSO dynamics. Here
we find a similar effect on multi-year timescales in response to an anomalous Aleutian Low.

The midlatitude westerly winds show a southerly shift throughout the year which, in agreement
with Liu et al. (2021), acts to prevent heat loss from the surface due to reduced evaporation. This
in turn drives the SST anomaly towards the equator. Liu et al. (2021) show the SFM as the
mechanism that propagates SST anomalies southward, through a change in latent heat fluxes.
However, in DJF the westerly winds imposed by the nudging cause a weakening of the subtropical
trades; hence the southerly shift of westerlies starts to occur within the season of nudging. We
show anomalous latent heat flux is responsible for the change in subtropical North Pacific SSTs.
The limitation of the Liu et al. (2021) study is that the atmosphere was coupled to a thermodynamic
slab-ocean, whereas we integrate a fully coupled ocean model allowing for a role of ocean
dynamical feedbacks. Sun and Okumura (2019) conducted a related investigation by imposing
heat flux anomalies associated with the North Pacific Oscillation, which is a coupled atmosphere-
ocean mode, but they imposed a fixed year round anomaly whereas the Aleutian Low shows
strongest variability in winter and therefore we only impose relaxation during boreal winter in our
experimental design.



In the tropical Pacific, the dominant mechanism responsible for the increase in SSTs is meridional advection, with the change to meridional current velocity driving the accelerated warming in boreal spring. This coincides with a northward cross-equatorial SST gradient and the development of an anomalous cross-equatorial southward pressure gradient. Cross-equatorial winds are generated, which, due to Coriolis force act to weaken the trades in the northern equatorial region, decreasing the surface latent heat flux and leading to a local warming. The heat budget analysis shows that surface heat fluxes are the primary warming agent during the nudging period, whereas a change to surface advection drives the warming in the central tropical Pacific. A comprehensive review of this mechanism, commonly referred to as the wind-evaporation-SST (WES) mechanism, is provided in Mahajan et al. (2008). Further, the mechanism has been posited as a pathway through which North Pacific SSTs can influence ENSO variability (Amaya et al. 2019). Investigation into equatorial thermocline depth shows a slight deepening of the thermocline in all seasons apart from SON, which is supported by changes in the vertical advection term (not shown). Figure 7 gives a pictorial representation of the combined mechanisms involved in translating the Aleutian Low anomaly into the deep tropics.

While the results make conceptual sense and are in broad agreement with studies using more comprehensive modelling tools (see earlier references), the amplitude of the response could be verified in other more detailed coupled climate models.

## 5. Conclusions

Externally-forced Aleutian Low trends have been implicated as a potential driver of recent variations in the Pacific Decadal Oscillation (Smith et al., 2016; Klavans et al., submitted). Here, we have investigated the potential influence of Aleutian Low trends on basin-wide low frequency Pacific sea surface temperature variability using nudging simulations in an intermediate complexity climate model. The target Aleutian Low state represents an extremely intense Aleutian Low state ($-3\sigma$ of winter monthly variability) applied during boreal winter. The intensified Aleutian Low induces a basin-wide SST response that resembles the model's internally-generated PDO with a comparable amplitude in the extratropics, but a substantially weaker amplitude in the equatorial Pacific by a factor of 4 to 5.

The findings presented here support that the PDO can, at least in part, be driven by remotely forced changes in the North Pacific atmospheric circulation independent of the tropics. However,



in our experiment the amplitude appears to be too weak to fully explain a multi-annual shift in the PDO. This suggests that the hypothesis posed by Smith et al. (2016) and Klavans et al. (submitted), that anthropogenically forced changes in the Aleutian Low drove the observed shift in the phase of the PDO in the late 20th and early 21st centuries, should be revisited.

**Code availability**

The nudging code used in the analysis can be found:
(https://github.com/NOC-MSM/FORTE2.0).

**Data availability**

Underlying model data found in this paper is available from the corresponding author upon request.

HadISST data available: https://www.metoffice.gov.uk/hadobs/hadisst/data/download.html

**Author contribution**

WJD and ACM designed the study. WJD developed the nudging code in FORTE2.0 with support from CMM, MMJ and RR. ATB and RR helped with installation of FORTE2.0 at Leeds. WJD performed the analysis and produced the figures. WJD and ACM wrote the manuscript with comments from all authors. All simulations were performed on the ARC4 HPC at the University of Leeds.

**Competing interests**

The authors declare that they have no conflict of interest.

**Acknowledgements**

WJD was supported by a Natural Environment Research Council (NERC) Ph.D. studentship through the SPHERES Doctoral Training Partnership (NE/L002574/1) and by a Met Office CASE studentship. ACM and CMM were supported by the European Union's Horizon 2020 Research



and Innovation Programme under Grant Agreement 820829 (CONSTRAIN project). ACM was
supported by the Leverhulme Trust. We are grateful to Paloma Trascasa-Castro for discussion of
ENSO processes. We are grateful for feedback on an earlier version of this manuscript from John
Marsham and Laura Wilcox.

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

**Figures**




**Figure 1:** Annual mean surface temperature anomalies for (a) regression onto the PDO index in CONTROL; (b) ensemble mean anomaly in NUDGED averaged over years 1-30. Units are K per standard deviation. Stippling denotes anomalies that are significant at the 95% level. Green and black boxes show the regions for the mixed layer heat budget analysis.

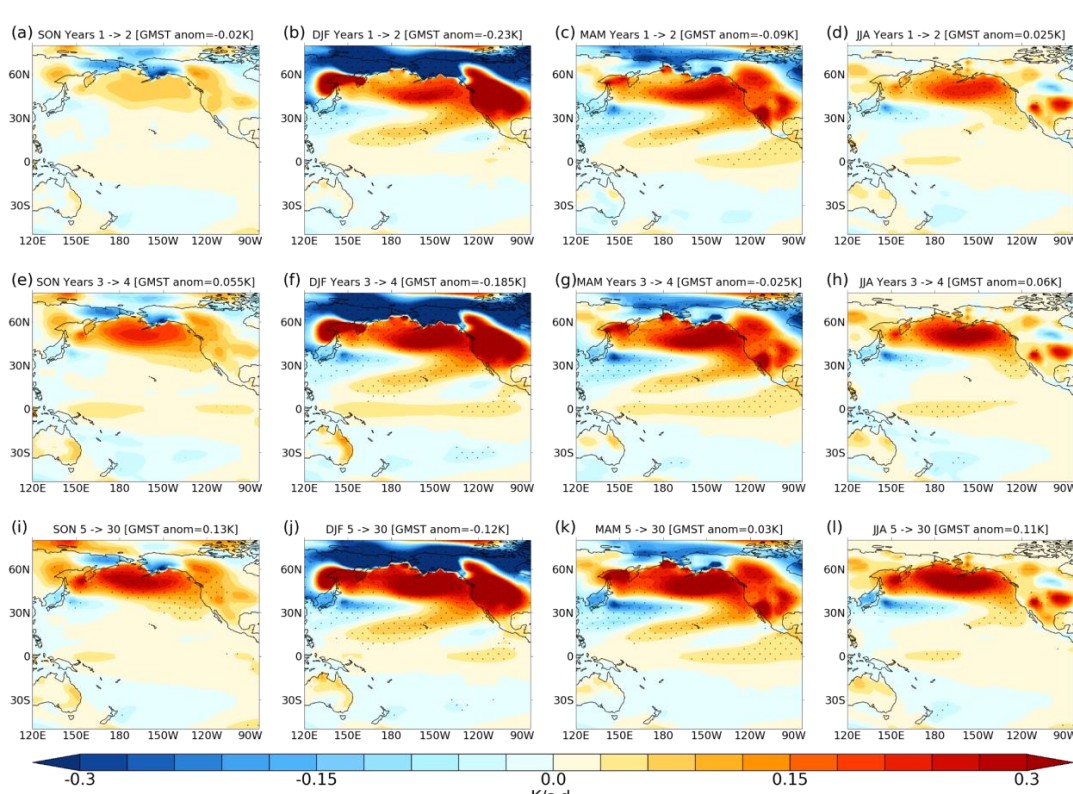

**Figure 2:** Seasonal mean surface temperature anomalies in NUDGED expressed per unit PDO index [K/σ] for SON, DJF, MAM and JJA. Anomalies are shown for years 1-2 (a-d), years 3-4 (e-h) and years 5-30 (i-l). Global mean surface temperature anomalies are shown in the header. Stippling denotes anomalies that are significant at the 95% level.











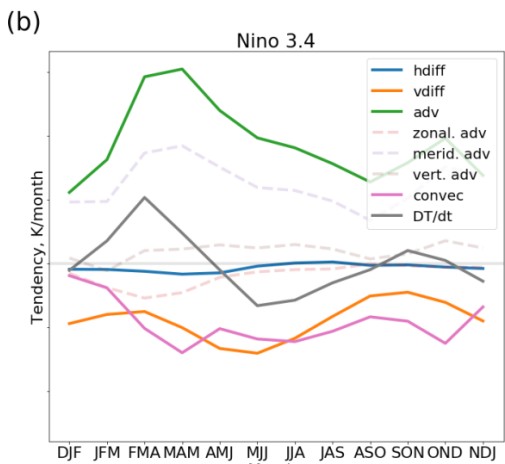




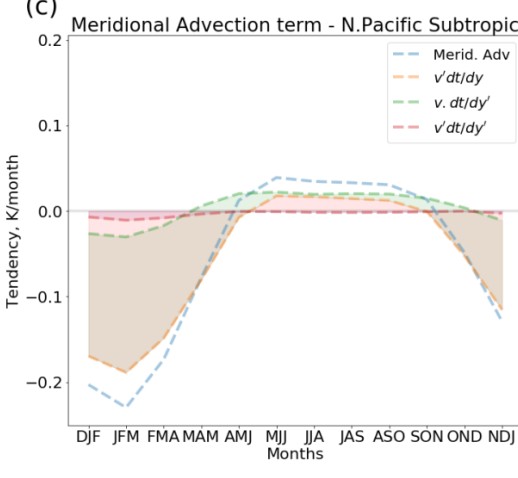

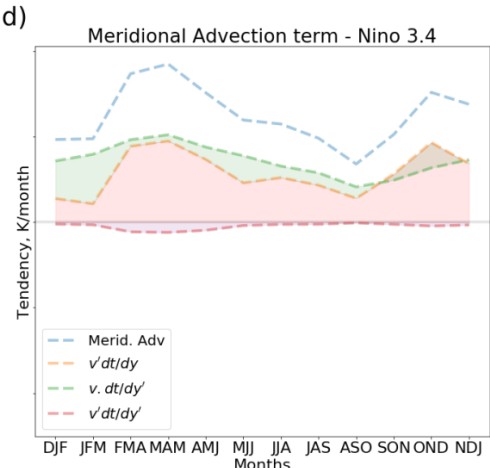





**Figure 3:** 3-month moving average of mixed layer temperature tendencies and
constituent heat budget terms for the (a) subtropical North Pacific and (b) Niño 3.4
regions. (c,d) show the meridional advection term and its linear expansion.

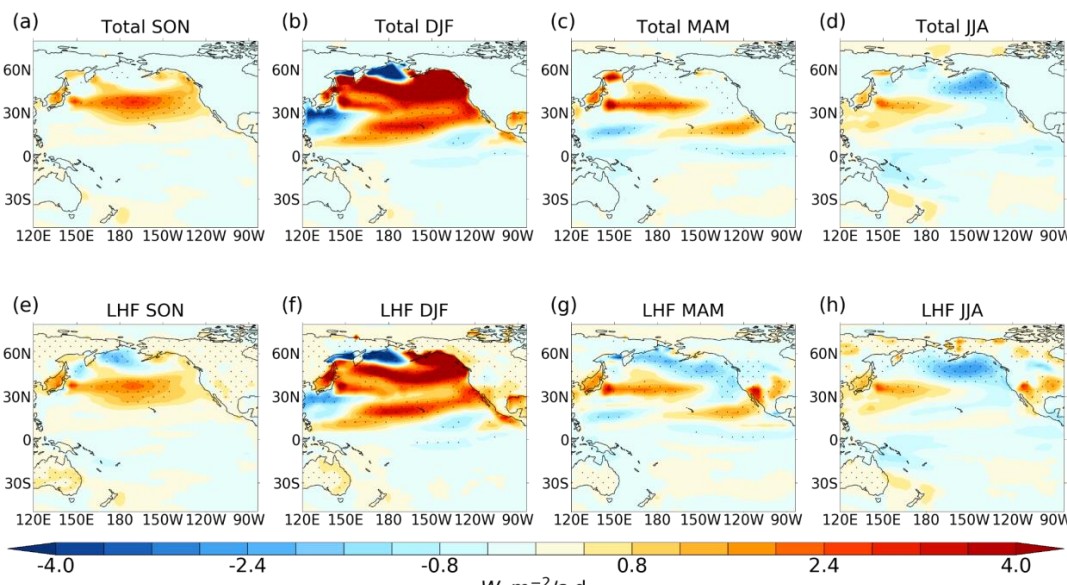


**Figure 4**: (a-d) Seasonal mean net surface heat flux anomalies in NUDGED. (e-h):
Seasonal mean latent heat flux anomaly in NUDGED. Positive denotes downward flux.
Stippling denotes anomalies that are statistically significant at the 95% level. Units: W m⁻
² per standard deviation.








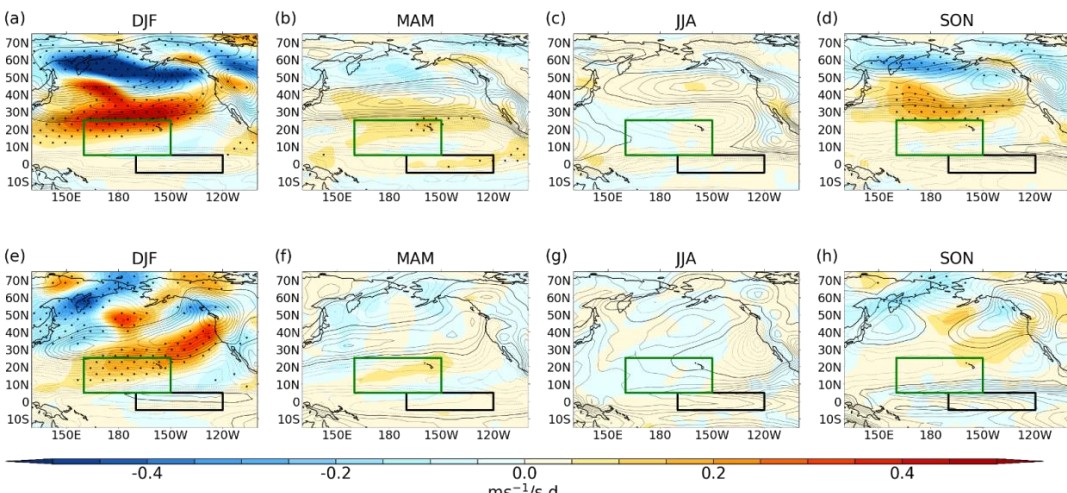

**Figure 5:** Seasonal mean NUDGED near-surface wind anomalies for (a-d) zonal and (e-h) meridional wind. Contours show climatology of CONTROL (dashed lines are negative values, contour interval 1 m s$^{-1}$).  Stippling denotes anomalies that are significant at the 95% level.





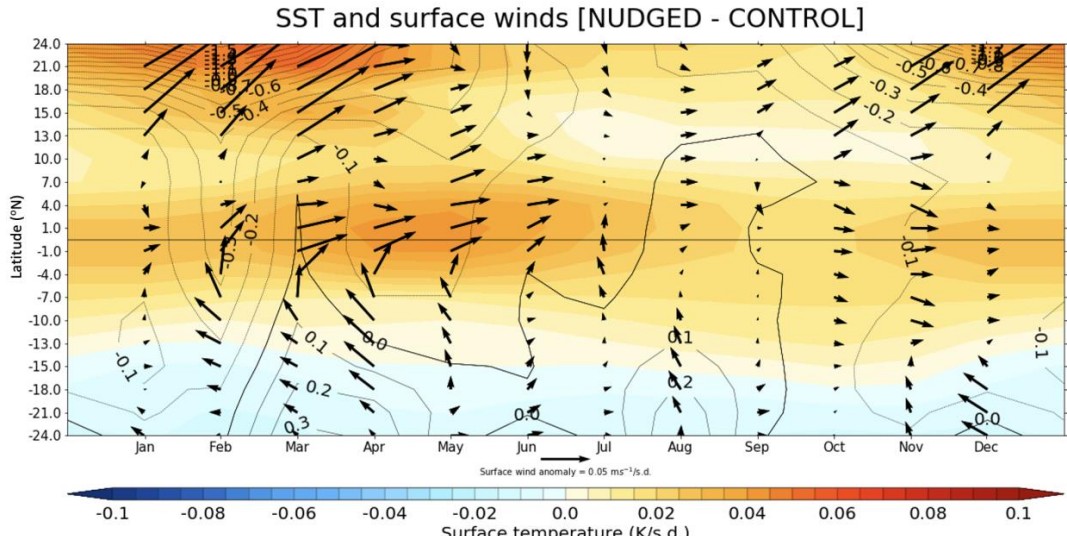

**Figure 6:** Latitude-time section of SST anomaly (K/σ: shading), surface pressure (hPa/σ: contours) and near-surface wind anomaly (m s$^{-1}$/σ: vectors) averaged over the central-eastern tropical Pacific (205ºW-80ºW).





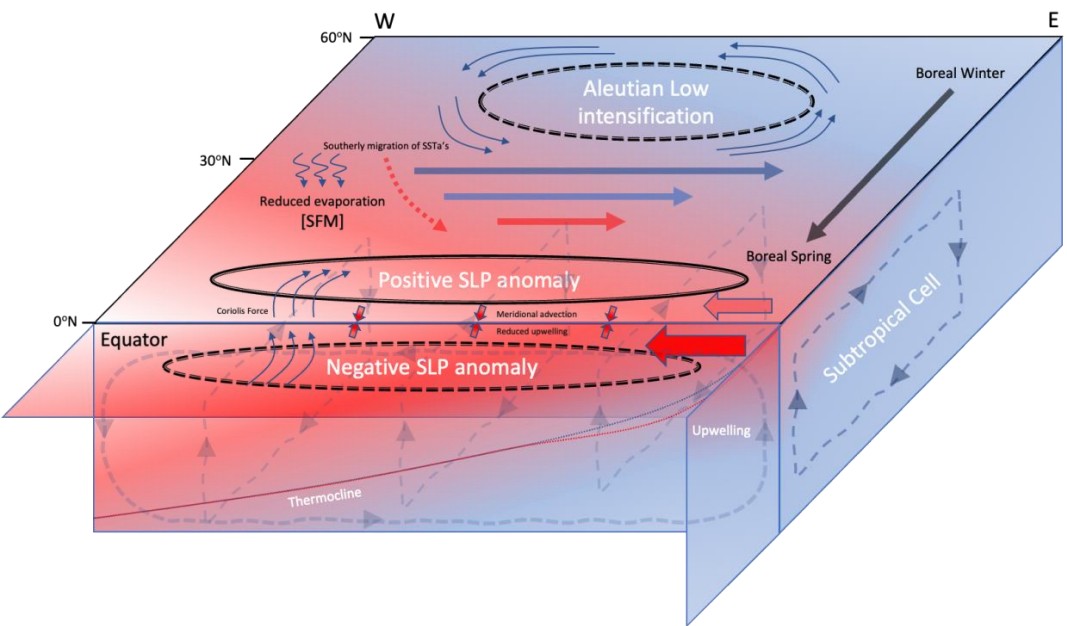



**Figure 7:** Schematic depicting the mechanisms involved in the tropical SST anomalies
manifest as a result from an intensification of the AL. An intensified AL (dashed black
line) imposed during boreal winter is associated with intensified westerlies (solid arrows)
in the extra-tropics and downward latent heat transfer. The migration of the SST
anomalies southward during boreal winter is associated with a southerly shift in the
westerly anomalies. The westerly anomalies act to weaken the background trades (filled
red arrows) which reduce latent heating due to evaporation and hence an increase in
extra-tropical Pacific SSTs. In the season after nudging, the temperature asymmetry
either side of the equator induces an SLP gradient (solid line – positive SLP; dashed
line – negative SLP) that drives southerly winds across the equator. The Coriolis force
acts to turn the southerly winds in the southern hemisphere westward and in the
northern hemisphere eastward. When these anomalous winds are imposed on the
background easterly trade winds (filled red arrows), the southerlies south of the equator
increase the wind speed and therefore evaporative cooling, whilst north of the equator
the background trades are weakened, reducing evaporative cooling. The changes to the
wind driven surface state act to deepen the thermocline in the eastern tropical Pacific
(red dotted line) and reduce upwelling/divergence of cooler waters at the equator.