# Peer review of "Intensified Aleutian Low induces weak Pacific Decadal Variability"

_EGUsphere, 2023_

## Editor Comment (EC1)

Dear Dr. Dow:

The comments by the two reviewers are pertinent and I hope you take them all into consideration in revising the manuscript. I have further comments that I would hope you address in revising the manuscript that should help the reader to get a clearer picture of what is going on in the model.

A basic calculation that should be presented in the revised manuscript is a comparison of the amplitude of the 3 sigma Aleutian Low anomaly in the model to the amplitude of an observed 3 sigma anomaly. That is, is the forcing applied to the model of realistic amplitude, or is the simulated variability in the Aleutian Low to weak? If the latter is the case, then that would explain why the observed PDO variability is too weak in the model. But I expect there is another, more likely, explanation for the "apparent" weakness of the PDO variability in the model compared to observations. (In fact, as far as I can tell by the figures in the manuscript, the SST variability driven by the internal intrinsic Aleutian Low variability is quite consistent with that observed.)

In reading the manuscript, it is clear that some of the concerns of the reviewers stems from a misinterpretation of what is the PDO. This paper uses an old definition of the PDO (the first EOF of SST in the N. Pacific; e.g., Mantua et al. 1997) that reflects the leading pattern of interannual to decadal variability in the N. Pacific. As such, it includes interannual SST variability that is driven by ENSO and the interannual and lower frequency SST variability that is driven by the internal variability in the Aleutian Low (as measured by the NPI index) and external forcing (e.g., volcanic eruptions). The SST patterns associated with both driving mechanisms are very similar. Hence, the traditional "PDO" index used in the current manuscript includes SST variability due to ENSO variability (hereafter $PDO_{hf}$) and SST variability due to stochastic forcing by the Aleutian Low that (hereafter $PDO_{lf}$) is intrinsic to the midlatitude atmosphere (i.e., associated with fixed climatological SST); hereafter I use quotes to denote the traditional PDO index, "PDO", because it is a statistical artifact that conflates two different driving processes). When statistical methods are used to remove the ENSO contribution from the "PDO", the extratropically driven contribution ($PDO_{lf}$) is well described as the response of the midlatitude ocean to stochastic forcing by the internal variability associated with the Aleutian Low, with SST anomalies local to the climatological Aleutian Low driven by turbulent heat fluxes, and delayed SST anomalies in the Kuroshio region driven by the ocean gyre adjustment to wind stress curl anomalies associated with a stochastic Aleutian Low  (Wills et al. 2019, Newman et al. 2016, Zhao et al. 2021 and references therein). This view that the PDO is an extratropical phenomenon stems from analysis of the observations and analysis of the CMIP5 and CMIP6 climate models (ibid), and is now widely referred to the PDO. Importantly, the Wills et al show the $PDO_{lf}$ has only a very weak footprint in the tropical Pacific.  (Zhao et al show secondary feedbacks that couple the tropical Pacific and midlatitude N. Pacific, contributing to the variance along the west coast of the North America).

There are several important implications for the interpretation of the results presented in the authors manuscript. First, ENSO contributes approximately equally to the observed "PDO" amplitude. Hence, the discrepancy in the amplitude of the "PDO" seen in Fig S1 may be due to an ENSO that is too weak in the model that is superimposed on a *realistic* amplitude $PDO_{lf}$ variability. Given that the observed $PDO_{lf}$ has a very weak footprint into the equatorial Pacific,

one would have to scale up the model "PDO" to see the weak tropical footprint of the PDO$_{lf}$ to match the PDO pattern in the observation. Indeed, it seems to me that if the amplitude of the midlatitude SST anomalies in Fig S1 in the control were divided by ~3, they would have the nearly the same amplitude of the PDO$_{lf}$ anomalies in the observations shown in Wills et al.[1] Second, if this is the case, then the present study is a consistent with the analysis of observations that show the SST variability driven by anomalies in the Aleutian Low have a very weak footprint in the equatorial Pacific.

Specific comments:
The title: it doesn't really make sense. *Variability* in the Aleutian Low provides the forcing for the SST *variability* in the N. Pacific on decadal and multidecadal time scales that features the PDO pattern of SST anomalies. What this paper has shown is that a *climatological* change (increase) in the amplitude of the Aleutian Low will act change the *climatological* SST with a pattern that is very similar PDV pattern; it cannot change the variance in the PDV, however. A more apt description of the paper conclusions is that a persistent positive Aleutian Low forcing causes a persistent PDO-like pattern of SST change in the N. Pacific.

Lines 57-58, "The prevailing paradigm for the PDO regards the role of the Aleutian Low to be largely driven by tropical processes". This is not true. The prevailing paradigm of the PDO is that is interannual to decadal variability in the North Pacific Ocean that is driven primarily by stochastic variability in the Aleutian Low that is largely unrelated to changes in SST – including changes in tropical Pacific SST; it includes SST anomalies driven by turbulent heat flux anomalies associated with the Aleutian Low anomalies and the delayed response due to the integrated wind stress curl anomalies associated with the Aleutian Low.    The leading pattern of SST variability in the North Pacific (the traditional PDO definition) includes both the midlatitude PDO and a contribution due to ENSO variability, communicated to the N. Pacific by atmospheric teleconnections.

Line 81: please add Wills et al 2019 to the list of references, as this brings the list up to date and arguably is the cleanest description of the modern day view of the PDO.

Line 135-136: Nudging is applied to the region of the observed climatological (DJF) Aleutian Low, but does it also align with the model simulated Aleutian Low? [See also comment #2 of reviewer #2.] Is the pattern and amplitude of the leading mode of atmospheric variability in the observations, the Aleutian Low/NPI index, consistent with that simulated by the model? Panels b and f in Fig. S1 suggest the model Aleutian Low/NPI might be centered ~30 degrees west of that observed.

The description of the heat budget analysis is confusing. It is well known that the turbulent fluxes are the leading term in the variance budget for winter averaged SST tendencies and they should explicitly appear in Eqn. 5 and not lumped in with diffusion. See also comments by reviewer 1 on the 30 m depth vs. mixed layer depth.
* * *
[1] To fix ideas, consider the SST anomalies along 40-50N in the central Pacific. A observed 1 sigma ENSO event (a 0.8C Nino3 anomaly) causes a ~0.4C anomaly (PDO$_{hf}$) in SST in this region. From Wills et al, a SST anomaly associated with a 1 sigma PDO$_{lf}$ is ~0.2C. If these were independent processes and shared the same pattern, then the standard deviation of the total variability in the extratropics would be 0.44C =sqrt(0.2^2 +0.4^2)). However, if the model had a very weak ENSO compared to observations (say 1 sigma Nino3 of only 0.37C, 1/3 of that observed) but a realistic PDO$_{lf}$, then the extratropical SST amplitude would be 0.24C (=sqrt((0.4/3)^2 +0.2^2)) and scaling the model result by a factor of 2 or 3 would bring the tropical and extratropical SSTs in line with observations.

The tropical anomalies in the schematic in Figure 7 and the description in the text doesn't make sense to me. Figure 6 shows that in response to the Aleutian Low forcing, warm anomalies and negative SLP anomalies in the northern subtropics ~20N, and cold SST anomalies and positive SLP anomalies in the southern subtropics ~20S; hence, a cross equatorial pressure gradient to the north (there must also be a zonal pressure gradient to accompany the zonal wind anomalies centered on the equator). The schematic shows just the opposite.

I concur with Reviewer #2 that the similarity of the patterns in Figs. 1a and 1b should be quantified by a pattern correlation.

In the captions to figures 2-6, please note the averaging period that is being displayed. Are these figures composites for years 1 to 2, years 3-4, years 5-30?

Figure S1 (bottom panels) show the observed PDO has a similar pattern with the same sign in all seasons. This makes sense because the SST is dominated by low frequency variability. It is difficult to explain how, in the model, SON differs can differ in sign from the other three seasons.

Please show the same field in the top and bottom rows of Fig S1 (presently, it appears that the simulated 2m temperature is shown in the top row, but the observed SST is shown in the bottom row. I suggest showing SST for both.

Figure S3 is confusing. The contours are the same in all four panels, yet the caption states that anomalies are contoured. The amplitude of the surface heat flux anomalies (~ 0.03 W/m$^2$) seems to be two orders of magnitude too small to explain the SST anomalies. Most confusing of all is the pattern of heat flux anomalies that accompany the imposed Aleutian Low anomaly: the pattern should look like that in Fig. 4b and 4f (and in observations) – and yet there are anomalies of opposing signs on along the southern flank of the imposed Aleutian Low.

References

Mantua, N., S. Hare, Y. Zhang, J. Wallace, and R. Francis, 1997: A Pacific interdecadal climate oscillation with impacts on salmon production. *Bulletin of the American Meteorological Society*, **78**, 1069–1079, https://doi.org/10.1175/1520-0477(1997)078<1069:APICOW>2.0.CO;2.

Newman, M., and Coauthors, 2016: The pacific decadal oscillation, revisited. *Journal of Climate*, **29**, 4399–4427, https://doi.org/10.1175/JCLI-D-15-0508.1.

Wills, R. C. J., D. S. Battisti, C. Proistosescu, L. Thompson, D. L. Hartmann, and K. C. Armour, 2019: Ocean Circulation Signatures of North Pacific Decadal Variability. *Geophysical Research Letters*, **46**, 1690–1701, https://doi.org/10.1029/2018GL080716.

Zhao, Y., M. Newman, A. Capotondi, E. D. Lorenzo, and D. Sun, 2021: Removing the Effects of Tropical Dynamics from North Pacific Climate Variability. *Journal of Climate*, **34**, 9249–9265, https://doi.org/10.1175/JCLI-D-21-0344.1.

---

## Editor Comment (EC2)

Dear Dr. Dow and co-authors:

Thank you for your thoughtful responses to the concerns and comments of the Reviewers on your manuscript. Reviewer #2 has remaining concerns that must be addressed before the paper can be accepted for publication in WCD (posted to you, I believe). Below I add my concerns which should be addressed, and some suggestions to make text. Line numbers refer to the uploaded pdf named "egusphere-2023-1595-manuscript-version3.pdf"

Regards, David

- In his original review, Reviewer 2 asked: "What does it mean to express the anomalies between NUDGED and CONTROL "per standard deviation of the PDO index"? And how does one compare the amplitudes between the two?" Your response (reproduced next in blue) was very helpful and I strongly urge you to include this in the text – and to add similar text to the caption of Figure 1.

  *The anomaly between NUDGED and CONTROL is projected onto the first EOF from the control run to generate a pseudo-PC. The anomaly is divided by the pseudo-PC to calculate the anomaly per standard deviation of the PDO index expressed in a similar way to that derived from CONTROL.*

- The schematic in Fig. 7. is somewhat misleading: The negative SLP anomaly north of the equator is centered near ~30N and not 10N as presently shown. This is important because the south of the maximum (near 10-20N) there will be anomalous westerlies – reduced trade winds, and thus reduced evaporation. Also, the red arrows just south of the equator that point northward are inconsistent with the SLP field (they should point southward). Finally, the text in the figure caption should be sharpened to avoid confusion (in particular, subtropics usually refers to ~10-30N, while extratropics includes the midlatitudes and subpolar regions). I suggest replacing the caption with the following (or something like it):

  "**Figure 7:** Schematic depicting the mechanisms involved in the tropical SST anomalies manifest as a result of an intensification of the AL. An intensified AL (dashed black line) imposed during boreal winter is associated with westerly anomalies (reduced easterlies; solid red arrows) in the subtropics and downward latent heat transfer. The migration of the SST anomalies southward during boreal winter is associated with westerly anomalies in the subtropics (reduced trades). The westerly anomalies act to weaken the background trades (filled red arrow) which reduces latent cooling due to decreased evaporation and hence an increase in subtropical Pacific SSTs. In the season after nudging, the temperature asymmetry about the equator induces an SLP gradient (solid black line, positive SLP; dashed black line, negative SLP) that drives southerly winds across the equator. The Coriolis force acts to turn the southerly winds in the southern hemisphere westward and in the northern hemisphere eastward. When these anomalous winds are imposed on the background easterly trade winds (filled red arrows), the southerlies south of the equator increase the wind speed and therefore evaporative cooling, whilst north of the equator the background trades are weakened, reducing evaporative cooling. The westerly wind anomalies along the equator

deepen the thermocline in the eastern tropical Pacific (red dotted line) and reduce upwelling/divergence of cooler waters at the equator."

- The paragraph on lines 73-93 is not relevant and is a distraction for the reader. Please remove it.
- There is some sloppiness in Eqns. 1-6 that need to be fixed. I will attach at the end of this document a page that will help.
- Line 154, change to read "within the nudging period (d = 0 is 15 Jan)".
- Line 158-162, this is confusing. How the amplitude of the imposed anomaly compares to the maximum amplitude in ERA5 isn't helpful. What is relevant is how the variability in the CONTROL compares to the variability in ERA5. With this in mind, I suggest you change the text on line 158 to read "… with an NPI anomaly of-10.76 hPa, or -3.02 $\sigma$, where $\sigma$ = 3.53 hPa is the standard deviation …" , change the text on line 161-2 to read "… reanalysis data from 1979-2020, a 1 $\sigma$ NPI anomaly is 5.20 hPa.", and change the text on line 163 to read "… conducted using a comparably sized NPI anomaly in reanalysis data."
- Line 223, "… Pacific Ocean …"
- Line 275, change to read "… There are positive (downward) …"
- Line 279-282, change this sentence to read "The pattern of surface latent heat flux anomalies in JJA in the extratropical North Pacific resembles the SST pattern associated with the internal PDO (Fig. S1d) and represents a damping of the SST anomalies; positive flux anomalies extend eastward from the KOE region, which are enveloped by negative anomalies in the northeast Pacific and subtropical North Pacific. The … "
- Line 299, change to read "…zonal wind anomalies represent a …"
- Line 331, change to read "…from the surface in the northern subtropics due to reduced…"
- Line 340-3421, the NPO is an intrinsic mode of atmosphere variability, not an intrinsic coupled atmosphere-ocean mode. Change to read "…the North Pacific Oscillation (NPO), but they imposed …"
- Line 350, change to read "…coincides with an anomalous northward…"
- Line 358, replace "Investigation into" with "The"
- Line 355, change to read "…the warming in the central near-equatorial Pacific…"
- Figure caption 3: change "NUDGED-CONTROL" to read "NUDGED minus CONTROL". Also, add the text "The subtropical North Pacific and Nino3.4 domains are indicated by the boxes in Fig. 1".
- Figure caption 6: change "NUDGED-CONTROL" to read "NUDGED minus CONTROL".
- Figure caption S1: change "Seasonal mean surface" to read "Seasonal mean skin"
- Figure caption S4: What are the box limits? The whisker limits? Presumably 10, 25 75 and 90%, but best to state this explicitly rather than making the reader guess. The text on line 36 in parentheses is confusing. Suggest replacing this text with the sentence: "The maximum and minimum values of Nino3.4 in the HadISST4 and Control run are indicated by an "x" (and then put the x's on the plot.

---

## Editor Comment (EC3)

**Edits on Equations 1-6**

December 29, 2023

The equations that appear in version 3 of the manuscript:

$$\delta x(\lambda, \phi, z, t) \;=\; -\gamma(\lambda, \phi, z, t)\,(x(\lambda, \phi, z, t) \;-\; x_{ref}(\lambda, \phi, z, t))\,/\tau \quad , \tag{1}$$

$$\gamma(\phi, \lambda) \;=\; f(\phi, \phi_1, \phi_2)\, f(\lambda, \lambda_1, \lambda_2) \quad , \tag{2}$$

$$f(\phi, \phi_1, \phi_2) \;=\; {}^{\iota}\left[ 1/(1 + e^{-(\phi-\phi_1)/\delta_1} \right]\; \left[ 1/(1 + e^{-(\phi-\phi_2)/\delta_2} \right] \tag{3}$$

$$f(\lambda, \lambda_1, \lambda_2) \;=\; {}^{\iota}\left[ 1/(1 + e^{-(\lambda-\lambda_1)/\delta_1} \right]\; \left[ 1/(1 + e^{-(\lambda-\lambda_2)/\delta_2} \right] \tag{4}$$

$$f(z) \;=\; a.\; exp(bx) \tag{5}$$

$$f(t) \;=\; \left( \frac{1}{exp\left(-0.5\left(\frac{d^2}{\beta^2}\right)\right)^{2\mu}} \right) \tag{6}$$

The problems with Eqns 1-4 are fixed by writing:

$$\delta x(\lambda, \phi, z, t) \;=\; -\gamma(\lambda, \phi)\, g(z)\, h(t)\,(x(\lambda, \phi, z, t) \;-\; x_{ref}(\lambda, \phi, z, t))\,/\tau \quad , \tag{1a}$$

$$\gamma(\lambda, \phi) \;=\; f_1(\phi, \phi_1, \phi_2)\, f_2(\lambda, \lambda_1, \lambda_2) \quad , \tag{2a}$$

$$f_1(\phi, \phi_1, \phi_2) \;=\; {}^{\iota}\left[ 1/(1 + e^{-(\phi-\phi_1)/\delta_1} \right]\; \left[ 1/(1 + e^{-(\phi-\phi_2)/\delta_2} \right] \tag{3a}$$

$$f_2(\lambda, \lambda_1, \lambda_2) \;=\; {}^{\iota}\left[ 1/(1 + e^{-(\lambda-\lambda_1)/\delta_1} \right]\; \left[ 1/(1 + e^{-(\lambda-\lambda_2)/\delta_2} \right] \tag{4a}$$

Eqn 5 doesn't align with Fig. S2: if $z$ is height above the surface (standard notation), then $f$ goes to infinity as you go upward. Is this what you mean to write here?

$$g(z) = a\,exp(-b\,z) \qquad (5a)$$

Note, the middle panel in Fig. S2 does not fit either description (the curve should go exponentially to 100% at "model level" = 1, but the figure displays a kink).

Eqn 6 doesn't align with Fig. S2 (as you go far from Jan 15, the denominator goes to zero and $f$ goes to infinity. $d$ appears to have units of month, but this isn't mentioned in the text. A more precise formulation would be

$$h(t) = exp\left(-d^2/\left(2\beta^2\right)^{2\mu}\right) \qquad (6a)$$

where $d$ is the time difference relative to maximum nudging time in months (e.g., $d = 0$ on Jan 15, $d = -1$ on Dec 15, etc). Outside of the nudging window, $h = 0$.

**Additional issues with these equations:**

- Eqns 3 and 4 don't seem to align with the mask shown in Fig. S2. Why are there two nodal points in latitude $(\lambda_1, \lambda_2)$ and longitude $(\phi_1, \phi_2)$, and what are their values? Also, $f_1$ and $f_2$ do not go to zero as you go far from the center of the patch. It seems like these equations should read as follows: "Within the nudging patch centered at $\lambda_1, \phi_1$,

$$f_1(\phi, \phi_1) = exp\left(-((\phi - \phi_1)/\delta_1)^2\right) \qquad (3a)$$

and

$$f_2(\lambda, \lambda_1) = exp\left(-((\lambda - \lambda_1)/\delta_2)^2\right) \quad . \qquad (4a)$$

and outside of the patch, $f_1 = f_2 = 0$. " Note that I am assuming you used a smooth function around the center of the patch (a Gaussian). If instead, you used the exponential (as suggested by Eqn. 3), $((\phi - \phi_1)/\delta_1)^2$ would be replaced with $|\phi - \phi_1|/\delta_1$ .

- In Eqns 3 and 4, $\delta_1$ and $\delta_2$ are not defined in the text).

- In Eqn 5, $x$ and $b$ are not defined in the text (also, presumably $x$ should be $z$).

- The mathematical expressions on lines 133-134 appear to have been scrambled when the text was converted to the pdf.

---

## Author Comment (AC1)

**Responses to editor and reviewer comments Dow et al., "Intensified Aleutian Low induces weak Pacific Decadal Variability" submitted to Weather and Climate Dynamics**

**Editor:** We thank the Editor for sourcing two detailed reviews of our manuscript, as well as for providing their own detailed comments. We are pleased that both reviewers find merit in our study and support the study being published in WCD after some revision. The comments are constructive and we have addressed the points raised as detailed below in blue.

**Editor comments**

The comments by the two reviewers are pertinent and I hope you take them all into consideration in revising the manuscript. I have further comments that I would hope you address in revising the manuscript that should help the reader to get a clearer picture of what is going on in the model.

A basic calculation that should be presented in the revised manuscript is a comparison of the amplitude of the 3 sigma Aleutian Low anomaly in the model to the amplitude of an observed 3 sigma anomaly. That is, is the forcing applied to the model of realistic amplitude, or is the simulated variability in the Aleutian Low too weak? If the latter is the case, then that would explain why the observed PDO variability is too weak in the model. But I expect there is another, more likely, explanation for the "apparent" weakness of the PDO variability in the model compared to observations. (In fact, as far as I can tell by the figures in the manuscript, the SST variability driven by the internal intrinsic Aleutian Low variability is quite consistent with that observed.)

Thanks for the suggestion. A comparison has been added to Section 2.2. This shows that the most extreme winter monthly NPI anomaly in ERA5 (1979-2020) is $-3.56\sigma$ corresponding to a -18.13 hPa anomaly. The $-3.02\sigma$ anomaly applied in FORTE2 corresponds to a -10.76 hPa anomaly. Therefore the forcing applied in the experiments is not as strong as an equivalent anomaly based on reanalysis data. This may contribute to a weaker response in the model as compared to observations. We have noted this in the methods and discussion.

Added:

"The strong Aleutian Low state is taken from a 100 year long control run (CONTROL) based on a winter month with an NPI anomaly of $-3.02\sigma$ (-10.76 hPa), where $\sigma$ is the standard deviation calculated over all winter months in CONTROL (Figure S3). Therefore, the target state represents an extreme intense Aleutian Low state as simulated in FORTE2.0. Comparing with ERA5 reanalysis data from 1979-2020, the most intense winter month has an NPI anomaly of $-3.56\sigma$ (-18.13 hPa). The imposed atmospheric forcing is therefore weaker than if an equivalent experiment was conducted using reanalysis data."

In reading the manuscript, it is clear that some of the concerns of the reviewers stems from a misinterpretation of what is the PDO. This paper uses an old definition of the PDO (the first EOF of SST in the N. Pacific; e.g., Mantua et al. 1997) that reflects the leading pattern of interannual to decadal variability in the N. Pacific. As such, it includes interannual SST variability that is driven by ENSO and the interannual and lower frequency SST variability that is driven by the internal variability in the Aleutian Low (as measured by the NPI index) and external forcing (e.g., volcanic eruptions). The SST patterns associated with both driving mechanisms are very similar. Hence, the traditional "PDO" index used in the current manuscript includes SST variability due to ENSO variability (hereafter $PDO_{hf}$) and SST variability due to stochastic forcing by the Aleutian Low that (hereafter $PDO_{lf}$) is intrinsic to the midlatitude atmosphere (i.e., associated with fixed climatological SST); hereafter I use quotes to denote the traditional PDO index, "PDO", because it is a statistical artifact that conflates two different driving processes). When statistical methods are used to remove the ENSO contribution from the "PDO", the extratropically driven contribution ($PDO_{lf}$) is well described as the response of the midlatitude ocean to stochastic forcing by the internal variability associated with the Aleutian Low, with SST anomalies local to the climatological Aleutian Low driven by turbulent heat fluxes, and delayed SST anomalies in the Kuroshio region driven by the ocean gyre adjustment to wind stress curl anomalies associated with a stochastic Aleutian Low (Wills et al. 2019, Newman et al. 2016, Zhao et al. 2021 and references therein). This view that the PDO is an extratropical phenomenon stems from analysis of the observations and analysis of the CMIP5 and $CMIP_{ti}$ climate models (ibid), and is now widely referred to the PDO. Importantly, the Wills et al show the $PDO_{lf}$ has only a very weak footprint in the tropical Pacific. (Zhao et al. show secondary feedbacks that couple the tropical Pacific and mid latitude N. Pacific, contributing to the variance along the west coast of North America).

We acknowledge the recent advances in interpreting what has long been defined as the "PDO". However, one of the motivations for our study was the hypothesis that recent anthropogenic aerosol trends contributed to the change in phase of the PDO in the late 1990s (Smith et al., 2016). This proposes that the significant cooling in the equatorial Pacific in the late 1990s/early 2000s may have been triggered by an anomalous Aleutian Low forced by changing Asian aerosol emissions. Specifically, in the caption for their Figure S8 they say "Changes in the Aleutian Low impact the north-easterly trade winds in the north Pacific, leading to a coupled Pacific basin-wide response including the PDO (which can be characterised by the difference between tropical and north Pacific temperatures)". The goal of our study was indeed to test whether the Aleutian Low can generate a basin-wide SST response as proposed in Smith et al. (2016).

Smith, D., Booth, B., Dunstone, N. *et al.* Role of volcanic and anthropogenic aerosols in the recent global surface warming slowdown. *Nature Clim Change* 6, 936–940 (2016). https://doi.org/10.1038/nclimate3058

There are several important implications for the interpretation of the results presented in the authors manuscript. First, ENSO contributes approximately equally to the observed "PDO" amplitude. Hence, the discrepancy in the amplitude of the "PDO" seen in Fig S1 may be due to

an ENSO that is too weak in the model that is superimposed on a realistic amplitude PDO$_{lf}$ variability. Given that the observed PDO$_{lf}$ has a very weak footprint into the equatorial Pacific, one would have to scale up the model "PDO" to see the weak tropical footprint of the PDO$_{lf}$ to match the PDO pattern in the observation. Indeed, it seems to me that if the amplitude of the midlatitude SST anomalies in Fig S1 in the control were divided by ~3, they would have the nearly the same amplitude of the PDO$_{lf}$ anomalies in the observations shown in Wills et al.[1] Second, if this is the case, then the present study is a consistent with the analysis of observations that show the SST variability driven by anomalies in the Aleutian Low have a very weak footprint in the equatorial Pacific.

Thank you for raising this important point. The modelled ENSO is slightly weaker than observed, though the bias is modest. Blaker et al. (2021) show ENSO SST anomalies reach a maximum of 1°C for the region 5°S–5°N, 160–100°W, and anomalies near to the coast of Central and South America reach 0.7°C.

To address the above two points, we have added the following text to section 2.4:

"Here we define the PDO using the common index based on the leading EOF of North Pacific SST variability. Wills et al. (2019) showed that the tropical Pacific SST anomalies associated with this index are predominantly related to high frequency (e.g., ENSO) SST variability, while the extratropical part is related to turbulent heat flux and wind stress anomalies associated with intrinsic Aleutian Low variability. The discrepancy between the modelled and observed SST anomalies associated with the PDO index in Figure S1 could be due to the slightly weaker than observed ENSO amplitude in the model by around 33% (Figure S4) (see also Blaker et al., 2021)."

Specific comments: The title: it doesn't really make sense. Variability in the Aleutian Low provides the forcing for the SST variability in the N. Pacific on decadal and multidecadal time scales that features the PDO pattern of SST anomalies. What this paper has shown is that a climatological change (increase) in the amplitude of the Aleutian Low will act to change the climatological SST with a pattern that is very similar PDV pattern; it cannot change the variance in the PDV, however. A more apt description of the paper conclusions is that a persistent positive Aleutian Low forcing causes a persistent PDO-like pattern of SST change in the N. Pacific.

We have changed the title of the manuscript taking into account this, as well as reviewer 2's helpful comment. The new title is "Sustained intensification of the Aleutian Low induces weak tropical Pacific sea surface warming"

Lines 57-58, "The prevailing paradigm for the PDO regards the role of the Aleutian Low to be largely driven by tropical processes". This is not true. The prevailing paradigm of the PDO is that is interannual to decadal variability in the North Pacific Ocean that is driven primarily by stochastic variability in the Aleutian Low that is largely unrelated to changes in SST – including changes in tropical Pacific SST; it includes SST anomalies driven by turbulent heat flux

anomalies associated with the Aleutian Low anomalies and the delayed response due to the integrated wind stress curl anomalies associated with the Aleutian Low. The leading pattern of SST variability in the North Pacific (the traditional PDO definition) includes both the midlatitude PDO and a contribution due to ENSO variability, communicated to the N. Pacific by atmospheric teleconnections.

This section has been changed to:
"The traditional paradigm for the PDO describes the integrated effect of mid-latitude stochastic variability, which induces SST anomalies through turbulent heat flux and wind stress curl anomalies, and driving from tropical processes (ENSO variability) via excitation of Rossby wave trains and tropical-extratropical teleconnections (Newman et al. 2016; Zhao et al. 2021; Vimont. 2005; Knutson and Manabe 1998; Jin 2001). We note that recent definitions separate low frequency PDO variability and show this is predominantly associated with stochastic extratropical atmospheric variability (i.e. the Aleutian Low) (Wills et al., 2018, 2019)."

Line 81: please add Wills et al 2019 to the list of references, as this brings the list up to date and arguably is the cleanest description of the modern day view of the PDO.

Done

Line 135-136: Nudging is applied to the region of the observed climatological (DJF) Aleutian Low, but does it also align with the model simulated Aleutian Low? [See also comment #2 of reviewer #2.] Is the pattern and amplitude of the leading mode of atmospheric variability in the observations, the Aleutian Low/NPI index, consistent with that simulated by the model? Panels b and f in Fig. S1 suggests the model Aleutian Low/NPI might be centered ~30 degrees west of that observed.

Throughout the model development stage, the spatial pattern of modeled Aleutian Low was tested and found to be located in the same region as that defined by the observed variability. Moreover, the specific Aleutian Low state towards which the model is nudged is encapsulated well by the limits defined by observational data. In addressing comment #2 from reviewer #2, a caveat is added into the Discussion section:

"The simulations presented use an anomalous Aleutian Low state taken from a single month (Figure S3). An area for future research is to impose a suite of varying Aleutian Low states with different spatial and temporal profiles to test the sensitivity of the responses described here to details of the imposed relaxation state."

The description of the heat budget analysis is confusing. It is well known that the turbulent fluxes are the leading term in the variance budget for winter averaged SST tendencies and they should explicitly appear in Eqn. 5 and not lumped in with diffusion. See also comments by reviewer 1 on the 30 m depth vs. mixed layer depth.

The lack of further granularity in the terms defining the heat budget was due to limitations with available model diagnostics. In addition to taking into account comments by reviewer 1, the start of section 2.3 has been altered to:

"The heat budget of the upper 30m of the ocean (representing the mixed layer) is analysed for the regions shown by the boxes in Figure 1, where the temperature tendency is given by:

dT/dt = ADV + DIFFvert + DIFFhoriz + CONV (Eqn. 7).

Daily tendencies due to advection (ADV), vertical and horizontal diffusion (DIFFvert and DIFFhoriz) and convection (CONV) are output from the model. Further granularity in the heat budget terms (e.g. turbulent fluxes) was not possible due to the limitated availability of diagnostics from the model."

*Footnote: To fix ideas, consider the SST anomalies along 40-50N in the central Pacific. An observed 1 sigma ENSO event (a 0.8C Nino3 anomaly) causes a ~0.4C anomaly (PDO$_{hf}$) in SST in this region. From Wills et al, a SST anomaly associated with a 1 sigma PDO$_{lf}$ is ~0.2C. If these were independent processes and shared the same pattern, then the standard deviation of the total variability in the extratropics would be 0.44C =sqrt(0.2^2 +0.4^2)). However, if the model had a very weak ENSO compared to observations (say 1 sigma Nino3 of only 0.37C, 1/3 of that observed) but a realistic PDO$_{lf}$, then the extratropical SST amplitude would be 0.24C (=sqrt((0.4/3)^2 +0.2^2)) and scaling the model result by a factor of 2 or 3 would bring the tropical and extratropical SSTs in line with observations.*

Thanks for pointing this out. As noted above, the modelled ENSO is slightly weaker than observed but not ⅓ of the amplitude of observations. The standard deviation of the Nino3.4 index is 0.62K in the model and 0.77K in HadISST observations, so the amplitude is around 80% of the observations. Using the scaling above would give sqrt((0.4 * 0.8)^2 +0.2^2) = 0.38K, so this alone does not explain the difference in total variability of the North Pacific SST. It is possible there are differences in ENSO teleconnections to the Pacific extratropics which would give a different scaling factor than ~0.4K for PDO$_{hf}$. Future work will investigate this.

The tropical anomalies in the schematic in Figure 7 and the description in the text doesn't make sense to me. Figure 6 shows that in response to the Aleutian Low forcing, warm anomalies and negative SLP anomalies in the northern subtropics ~20N, and cold SST anomalies and positive SLP anomalies in the southern subtropics ~20S; hence, a cross equatorial pressure gradient to the north (there must also be a zonal pressure gradient to accompany the zonal wind anomalies centered on the equator). The schematic shows just the opposite.

Thanks for pointing this out. The schematic has been corrected.

I concur with Reviewer #2 that the similarity of the patterns in Figs. 1a and 1b should be quantified by a pattern correlation.

Done. Have added to Section 3.1:

"Across the Pacific ocean, the pattern of temperature anomalies in NUDGED closely resembles unforced multidecadal Pacific variability in CONTROL (Figure 1b), with a pattern correlation coefficient of 0.53."

In the captions to figures 2-6, please note the averaging period that is being displayed. Are these figures composites for years 1 to 2, years 3-4, years 5-30?

We have now stated the period used in each figure in the captions.

Figure S1 (bottom panels) show the observed PDO has a similar pattern with the same sign in all seasons. This makes sense because the SST is dominated by low frequency variability. It is difficult to explain how, in the model, SON differs in sign from the other three seasons.

This was an error in the plotting and has been fixed.

Please show the same field in the top and bottom rows of Fig S1 (presently, it appears that the simulated 2m temperature is shown in the top row, but the observed SST is shown in the bottom row. I suggest showing SST for both.

The top row shows surface skin temperature (not simulated 2m temperature), therefore is comparable with the bottom row.

Figure S3 is confusing. The contours are the same in all four panels, yet the caption states that anomalies are contoured. The amplitude of the surface heat flux anomalies (~ 0.03 W/m2) seems to be two orders of magnitude too small to explain the SST anomalies. Most confusing of all is the pattern of heat flux anomalies that accompany the imposed Aleutian Low anomaly: the pattern should look like that in Fig. 4b and 4f (and in observations) – and yet there are anomalies of opposing signs along the southern flank of the imposed Aleutian Low.

Thanks for pointing this out. The caption was incorrect - the contours show NDJFM SLP regressed onto the NPI index and are therefore identical in all panels. Furthermore, this figure has since been removed due to its lack of utility in drawing comparisons between regressions onto PDO and NPI indices.

References:

Mantua, N., S. Hare, Y. Zhang, J. Wallace, and R. Francis, 1997: A Pacific interdecadal climate oscillation with impacts on salmon production. Bulle'n of the American Meteorological Society, 78, 10ti9–1079, hCps://doi.org/10.1175/1520- 0477(1997)078<10ti9:APICOW>2.0.CO;2.

Newman, M., and Coauthors, 201ti: The pacific decadal oscillation, revisited. Journal of Climate, 29, 4399–4427, hCps://doi.org/10.1175/JCLI-D-15-0508.1.

Wills, R. C. J., D. S. Bawsti, C. Proistosescu, L. Thompson, D. L. Hartmann, and K. C. Armour, 2019: Ocean Circulation Signatures of North Pacific Decadal Variability. Geophysical Research Le@ers, 4ti, 1ti90–1701, hCps://doi.org/10.1029/2018GL08071ti.

Zhao, Y., M. Newman, A. Capotondi, E. D. Lorenzo, and D. Sun, 2021: Removing the Effects of Tropical Dynamics from North Pacific Climate Variability. Journal of Climate, 34, 9249– 92ti5, hCps://doi.org/10.1175/JCLI-D-21-0344.1.

**Reviewer #1:**

This paper examines the response of the Pacific climate system to forcing associated with Aleutian low variability using an intermediate complexity coupled atmosphere-ocean model (e.g., T42 atmospheric resolution; 2° ocean). The experiment design involves first performing a long control simulation. Nudging was used to insert atmosphere forcing associated with the Aleutian low, which is: i) applied between 30°-65°N 160°E-140°W; ii) begins in November, ramps up to a maximum between December-February and then ramps down in March; iii) is obtained from the average SLP anomaly over the N. Pacific (North Pacific Index) for when the NPI < -3σ in winter months (a very strong low pressure anomaly); iv) is full strength at the surface and decreases exponentially to zero at the tropopause; and v) added to temperature, winds and surface pressure on a daily basis. A 50-member nudged ensemble was performed, where each member is 30 years long and the nudging is applied each winter. The response to the nudged forcing is explored by the ensemble mean difference between the nudged ensemble and the control run.

The results suggest that the influence of the forcing can extend beyond the North Pacific to the subtropics and to a lesser extent to the equator. A regional mixed layer (upper 30 m) heat budget indicates that the temperature changes are mainly through the surface heat flux in the subtropical region, but meridional transport in the equatorial region. The Aleutian low was also identified as the primary driver of the Pacific decadal oscillation (PDO) SST pattern.

We thank the reviewer for their comprehensive comments and suggestions on the manuscript. We addressed the points raised below.

Comments

I have a few major concerns about the paper and some additional minor comments

Major comments:

1) A major issue of the results is the low amplitude of SST anomalies in general. The authors note that the Pacific low frequency variability in the control run is a factor of four-five times weaker than observations. In addition, the diffusivity between 5°N-5°S is increased by a factor of 20 to balance upwelling. While the model probably runs rapidly and so a large number of fairly-long simulations can be performed, is it really a good model for addressing the topic they wish to explore in this study? Perhaps statistical analysis could be applied to higher resolution GCMs, such as those in the CMIP archive, to investigate the relationships examined here. There are statistical methods, e.g., partial correlations, to examine correlation between variables while removing the influence of others.

We agree that relevant insight can be gained from models with varying complexities. FORTE2 has been extensively evaluated in the model documentation paper (Blaker et al., 2020) and employed in a recent study investigating another aspect of multi-decadal coupled atmosphere-ocean climate variability (Joshi et al., 2023). Therefore it is not unprecedented to

use a model of this kind to investigate low frequency climate variability and the flexibility of the model code enabled the nudging experiments to be performed. A caveat has been added to the end of the Discussion to acknowledge the potential limitations of the model and highlighting further work analysing existing more comprehensive models would be valuable.

"The coarseness of the coupled model, specifically the vertical dimension of the oceanic component, is a limitation of the study. Specifically, the model's relatively low resolution and inability to resolve mesoscale processes in the ocean and atmosphere may affect the results of the study. Future studies using observations and higher resolution GCMs to test the results herein would be valuable."

Joshi, M., Hall, R. A., Stevens, D. P., and Hawkins, E.: The modelled climatic response to the 18.6-year lunar nodal cycle and its role in decadal temperature trends, Earth Syst. Dynam., 14, 443–455, https://doi.org/10.5194/esd-14-443-2023, 2023.

2) Related to 1) and perhaps of even greater concern, is the very weak response to the forcing even though it's exceptionally strong (3σ NPI index) and applied over multiple winters. Many of the results are shown as a regression per unit change in the PDO σ (so it's a little difficult to gauge their actual values) but the values given are generally less than 0.1 K/σ south of 30°N and even smaller on the equator. Indeed lines 187-188 and Fig. 6 indicate that the maximum response in the equatorial band is less than 0.05 K/σ for all months. This is very small even for decadal variations along the equator. While the use of a large number of ensembles may indicate statistical significance in some subtropical locations, many fewer grid points have significant changes south of ~10°N. Even if the changes are significant due to the large sample size, they may not have much practical importance given their small magnitude. If anything, these results suggest that fluctuations in the Aleutian low have little impact on the equatorial Pacific (in contrast to forcing by the North Pacific Oscillation).

The authors agree that the magnitude of the effect in the tropics is minimal/weak and this has been acknowledged throughout the manuscript and explicitly in the title. However, as highlighted by the Editor this finding is broadly consistent with Wills et al. (2019) who statistically separate the low frequency component of the PDO and show this has weak tropical amplitude. It is, however, not consistent with the interpretation of some other studies who hypothesise a larger extratropical-driven signature of the PDO in tropical Pacific SSTs when using a traditional EOF-based PDO index (e.g., Smith et al., 2016; Klavans et al. submitted).

3) Given that the forcing is drawn from a coupled control experiment, the Aleutian low variability and the PDO obtained from the control run will include the influence of the tropics (via the atmospheric bridge).

The authors agree that the North Pacific variability in the control run will contain effects from tropical variability via the atmospheric bridge - we have added a sentence to the manuscript stating this explicitly. The month chosen for the nudging reference state coincides with an ENSO state with an index of 0.55. We have also quantified in the text the amplitude of the NPI anomaly

used in the nudging and compared this to an equivalent anomaly in ERA5 (see response to Editor comment 1).

"Furthermore, to ensure model stability, the anomalous nudging state was drawn from the coupled atmosphere-ocean control simulation. The Aleutian Low variability sampled from this simulation therefore includes effects from tropical variability. The month used as the reference state for the nudging coincides with an ENSO state (magnitude = 0.55) in the tropical Pacific. Further study could investigate more idealised AL states and their effects on extra-tropical-tropical communication."

4) While the budget analysis is helpful to understand the processes involved in the anomalies reaching the subtropics and equator, further analysis could help elucidate how the anomalies reach the subtropics and equatorial regions.  For example is there propagation of the signal south (and usually westward) associated with WES?  Is the wind anomalies consistent with the "trade wind charging hypothesis"?

Thanks for raising this interesting point. The wind anomalies do not show similarity with that described by the trade wind hypothesis. Figure R1 below shows the evolution of the seasonal wind anomalies along with the upper ocean velocities. Westerly wind anomalies are evident across the tropical Pacific as opposed to easterly anomalies found under trade wind charging conditions (Chakravorty et al. 2020). The authors agree that further analysis would investigate the attribution of these anomalous regions.

[Figure]

Figure R1: Left: Seasonal mean northward velocity anomaly in upper-most ocean layer. Right: Seasonal upward velocity anomaly in the upper-most ocean layer. Vectors denote anomalous surface winds.

Chakravorty, S., Perez, R. C., Anderson, B.T., Giese, B., Larson, S., & Pivotti, V., 2020. Testing the trade wind charging mechanism and its influence on ENSO variability. Journal of Climate. doi:10.1175/JCLI-D-19-0727.1

**Minor comments**:

1) Line 65 Could reference the following paper as well:

Gan, B. L. Wu, F Jia, S. Li, W. Cai, H. Nakamura, M. A. Alexander, and A. J. Miller, 2017: On the response of the Aleutian Low to greenhouse warming. J. Climate, 30, 3907-3925, doi: 10.1175/JCLI-D-15-0789.1

Added reference. Done

2) Fig. 1:

a) The caption and legend above the figures don't seem to match.

Thanks for pointing this out. Changed

b) The forcing is very different than the control at high latitudes – perhaps not unexpected as the surface temperatures may not be that strongly related to the PDO. It is also somewhat surprising that the negative air temperature anomalies near ~35°N are not of larger amplitude and extend further from Asia into the central/eastern Pacific in the nudging experiment.

Agreed. This is down to the nudging spatial filter, where the coefficients ramp down from 1 to 0 across Japan/Asia (see Figure S2) and are therefore weaker than those imposed across the central North Pacific.

3) Lines: 151-152 variability is calculated by multiplying the standard deviation of overlapping 15-year means by √2.

Explain why 15 years (some estimate of decadal variability)? Why multiply by sqrt of 2 (for statistical significance)?

This section has been rewritten:

"Statistical significance is defined by comparing the responses to the magnitude of simulated unforced decadal variability. At each grid point, overlapping 15-year mean anomalies are calculated from CONTROL. A 15-year time window was chosen to adequately capture decadal internal variability. The standard deviation of the mean anomalies from CONTROL was multiplied by square root of 2 to account for the fact that the variability of a difference in means is of interest. This estimates the variation of the difference in standard deviation between two independent averages, which have the same variance, that would be expected due to internal variability. The median value of the standard deviations is used and we show 95% significance as where the response value lies outside of the bounds 1.96 times the median standard deviation. This is similar to the method used in IPCC AR5 (2013)."

4) Line 156 The mixed layer varies with season and is generally deeper than 30 m especially in winter at higher latitudes, so perhaps it should just be stated as the budget of the upper 30 m rather than over the mixed layer. While it is likely a secondary factor (especially for a fixed depth budget), is penetrating solar radiation out the base of the mixed layer (30 m) considered in equation 5?

The text has been changed to emphasize that it is only the upper 30m which are considered for the analysis.

"The heat budget of the upper 30m of the ocean (representing the mixed layer) is analysed for the regions shown by the boxes in Figure 1, where the temperature tendency is given by…"

5) Line 178, It might be helpful to show the calendar sigma values for the PDO in the Control and compare them to observations. It could be shown in the supplemental.

This comment is difficult to understand. Figure S1 shows the comparison of the magnitude of the PDO in the control run against observations across seasons.

6) Lines 188-191 Since the response to tropical SST anomalies strongly influences the Aleutian low and the PDO, the basin-wide SST in the Control and included in the nudging would not be solely due to "internally generated coupled variability".

The statement the reviewer is referring to is a comment on the resemblance between the basin-wide SST pattern in the CONTROL experiment and that when forced by an anomalous AL.

We have edited the sentence to "Therefore, a sustained increase in Aleutian Low strength forces a basin-wide SST response which resembles that associated with internally-generated coupled variability in CONTROL.."

7) Lines 240-242 and Fig. 4. It looks like fluxes damp the SST anomalies during MAM and JJA over most of the Pacific, as expected from a stochastic model perspective (linear damping of the SST anomalies) and may indicate little dynamic feedback on the atmosphere.

Agreed. Have added:

"Regions such as those in the north-east North Pacific appear to dampen the SST anomalies during MAM and JJA, which may indicate limited dynamic feedback to the atmosphere. However, across the central North Pacific, the persistence of surface latent flux anomalies year-round is expected given the surface temperature persistence and alludes to ocean-atmosphere feedbacks."

8) The regression of the latent heat flux on the SLP over the North Pacific the (NPI index, Fig. S3) looks very different than the nudged experiment, even north of 30°N in the winter, when it would be expected that they would be fairly similar.

Agreed. However, one must bear in mind that, by construction, the variability of the AL and its associated signature in surface fluxes in CONTROL will not exactly resemble the extreme month used for NUDGED. The nudged experiment will only likely resemble that particular month of that year on which the input is based.

9) Line 250; Fig. 5. Do winds (shading) include the nudged forcing? If so, should the wind forcing be removed to see the response? Otherwise it is primarily showing the forcing over the North Pacific.

Yes, the winds include the nudged forcing. As the forcing is only applied across the winter months for consistency and to avoid confusion it was decided to show the full field across all seasons.

It's also difficult to see a Rossby wave response to the forcing based on the surface winds. Perhaps showing upper level geopotential or stream function would be helpful in this regard.

We have added a figure to the supplement (Fig. S5) showing the upper tropospheric winds and have amended the text:

"The meridional wind shows alternating southerly-northerly anomalies across the North Pacific orientated with a north-easterly tilt suggesting that a persistently strong AL invokes a modulation of the climatological Rossby wave train providing a pathway for atmospheric communication between the North Pacific and eastern tropical Pacific. Evidence for the modulation of the Rossby wave train is further evident in the upper tropospheric winds (Figure S5)."

Note, no significant change occurs in the tropical box except for two points in MAM (and the values are very small).

10) Line 253-255 The zonal bands of wind anomalies that reach the equator in the central Pacific extend from southwestward from Central America not from California.

Changed

11) Line 289-293. While the authors note the difference in the timing of the forcing and the type of forcing (fluxes vs. air temperature and winds) between their study and the one by Sun and Okumura (2019), a key difference is that the latter derived the forcing from the NPO as opposed to the Aleutian low. The former has a dipole pattern with the southern lobe being closer to the equator and thus may be more effective at influencing the tropics.

Thanks for pointing this out. Reference to the dipole structure of the NPO has been added.

**Reviewer 2:**

This study investigates the influence of the Aleutian Low on tropical Pacific sea-surface temperatures (SSTs), using wind-nudging experiments in an intermediate complexity climate model. The pathway the authors investigate has been proposed as a key part of the Pacific Decadal Oscillation (PDO) and is critical piece in understanding how tropical Pacific SSTs are influenced by internally generated or externally forced changes in the Aleutian Low. The results of this study are therefore potentially relevant for interpreting historical variability in Pacific SSTs and will be of broad interest within the climate variability community.

We thank the reviewer for their interest in our study and for making detailed and constructive comments. We address the points raised below.

While the overall setup of this study is mostly sound, the interpretation of the results is misleading in several substantial ways, which should be addressed before publication:

1) One important caveat that needs to be mentioned is that this model uses very low resolution that does not resolve mesoscale processes in the ocean and atmosphere. While this is true of many studies on the PDO, this is important here considering the framing in relation to Klavans et al. (submitted), who propose higher resolution as potentially helping to address the low signal-to-noise ratio of the PDO.

The authors agree with this point and have added at the end of the Discussion:

"The coarseness of the coupled model, specifically the vertical dimension of the oceanic component, is a limitation of the study. Specifically, the model's relatively low resolution and inability to resolve mesoscale processes in the ocean and atmosphere may affect the results of the study. Future studies using observations and higher resolution GCMs to test the results herein would be valuable."

*2)* Another important caveat that needs to be mentioned is that the nudging used is taken from a single winter and therefore may not be representative of Aleutian Low variability in general. It would be possible to do more analysis to show that the anomaly used is (or is not) representative of other 3-sigma Aleutian Low anomalies, which could get around needing to include this caveat.

This caveat is acknowledged in the Discussion by adding:

"The simulations presented use an anomalous Aleutian Low state taken from a single month (Figure S3). An area for future research is to impose a suite of varying Aleutian Low states with different spatial and temporal profiles to test the sensitivity of the responses described here to details of the imposed relaxation state."

*3)* Some of the framing of the results is about their relevance for decadal variability, e.g., the sentence "Here we find a similar effect on multi-year timescales in response to an anomalous Aleutian Low." However, the same nudging is imposed every year, so the results could entirely be explained by processes on shorter timescales. Please revisit each instance of 'decadal' and 'PDO' in the manuscript and reconsider whether these statements are supported considering the annually repeating forcing used. It is of course fine to motivate the study by questions about decadal variability, but considering the model set up, it's not possible to make strong conclusions about decadal variability without further work. The title and abstract are important to revisit in this regard. In my opinion, the abstract is well written and supported by the study up until "anomalous Pacific SST" on line 29, but after that all the statements about the PDO are only marginally related to the study and can't really be supported by the results. I also think that the inclusion of "Pacific decadal variability" in the title does not accurately represent what the study is about.

Thanks for this interesting comment. The abstract and title have been amended as suggested by the reviewer. Figure 2 shows the temporal evolution of the SST response to the Aleutian Low anomaly, which allows us to assess whether the long-term response is fully developed on interannual timescales. This reveals time dependence in both extratropical and tropical Pacific SST in response to the fixed Aleutian Low forcing, suggesting that multi-annual feedbacks are important for shaping the long-term quasi-equilibrium SST response. While in the real world the extratropical atmospheric forcing is stochastic, we believe we can still learn about the relevant feedbacks that shape the response in an idealised framework with fixed forcing.

4) On the influence of the Aleutian Low not being able to explain the full PDO nor it's phase shifts in the late 20th and 21st centuries, please consider recent literature showing that the decadal variability of the PDO, including its characteristic phase shifts, is coming almost entirely from the North Pacific (Wills et al. 2018, https://doi.org/10.1002/2017GL076327; Wills et al. 2019, https://doi.org/10.1029/2018GL080716), with the tropical Pacific primarily adding noise on interannual timescales. At the very least, please change "PDO" to "PDO in the tropics" on line 30. However, this also means that the statements on lines 30-31 and 329-332 are not supported by the results without additional analyses.

These references have been cited in the Introduction and the "PDO" terms have been refined in the abstract and throughout the manuscript.

Revised abstract:

"It has been proposed that externally forced trends in the Aleutian Low can induce a basin-wide Pacific SST response that projects onto the pattern of the Pacific Decadal Oscillation (PDO). To investigate this hypothesis, we apply local atmospheric nudging in an intermediate complexity climate model to isolate the effects of an intensified winter Aleutian Low sustained over several decades. An intensification of the Aleutian Low produces a basin-wide SST response with a similar pattern to the model's internally-generated PDO. The amplitude of the SST response in

the North Pacific is comparable to the PDO, but in the tropics and southern subtropics the anomalies induced by the imposed Aleutian Low anomaly are a factor of 3 weaker than for the internally-generated PDO. The tropical Pacific warming peaks in boreal spring, though anomalies persist year-round. A heat budget analysis shows the northern subtropical Pacific SST response is predominantly driven by anomalous surface turbulent heat fluxes in boreal winter, while in the equatorial Pacific the response is mainly due to meridional heat advection in boreal spring. The propagation of anomalies from the extratropics to the tropics can be explained by the seasonal footprinting mechanism, involving the wind-evaporation-SST feedback. The results show that low frequency variability and trends in the Aleutian Low could contribute to basin-wide anomalous Pacific SST, but the magnitude of the effect in the tropical Pacific, even for the extreme Aleutian Low forcing applied here, is small. Therefore, external forcing of the Aleutian Low is unlikely to account for observed decadal SST trends in the tropical Pacific in the late 20th and early-21st centuries."

5) More interpretation is needed in Section 3.3 (Figures 5 and 6). What are the take away's from this analysis?

Thanks for the comment. The authors have added more details to this section.

"The meridional wind shows alternating southerly-northerly anomalies across the North Pacific orientated with a north-easterly tilt suggesting that a persistently strong AL invokes a modulation of the climatological Rossby wave train providing a pathway for atmospheric communication between the North Pacific and eastern tropical Pacific. Evidence for the modulation of the Rossby wave train is further evident in the upper tropospheric winds (Figure S5)."

**Minor comments:**

For consistency between Eq. 1 and 2, I would recommend defining functions of z and t in Eq. 2 (as described near the bottom of page 5 / top of page 6).

Included as equations 5 and 6

Line 152: "multiplying the standard deviation of overlapping 15-year means by $\sqrt{2}$" requires more explanation of why you are doing this / why this is the relevant measure for statistical significance.

*This section has been rewritten:*

"'Statistical significance is defined by comparing the responses to the magnitude of simulated unforced decadal variability. At each grid point, overlapping 15-year mean anomalies are calculated from CONTROL. A 15-year time window was chosen to adequately capture decadal internal variability. The standard deviation of the mean anomalies from CONTROL was multiplied by square root of 2 to account for the fact that the variability of a difference in means is of interest. This estimates the variation of the difference in standard deviation between two independent averages, which have the same variance, that would be expected

due to internal variability. The median value of the standard deviations is used and we show 95% significance as where the response value lies outside of the bounds 1.96 times the median standard deviation. This is similar to the method used in IPCC AR5 (2013)."

I think (a) and (b) are switched in the caption of Figure 1

Corrected

Line 183: What does it mean to express the anomalies between NUDGED and CONTROL "per standard deviation of the PDO index"? And how does one compare the amplitudes between the two (as discussed on lines 195-196).

The anomaly between NUDGED and CONTROL is projected onto the first EOF from the control run to generate a pseudo-PC. The anomaly is divided by the pseudo-PC to calculate the anomaly per standard deviation of the PDO index expressed in a similar way to that derived from CONTROL.

Line 188-189: Considering that Fig. 1a and Fig. 1b DO NOT look very similar, please also discuss the differences, rather than just saying these patterns closely resemble one another. You could also make this more quantitative with a pattern correlation.

Agreed - have added this into the description of the results.

"Across the Pacific ocean, the pattern of temperature anomalies in NUDGED closely resembles unforced multidecadal Pacific variability in CONTROL (Figure 1b), with a pattern correlation coefficient of 0.53. Therefore, a sustained increase in Aleutian Low strength forces a basin-wide SST response which resembles that associated with internally-generated coupled variability in CONTROL. However, there are clear differences in the sign of the anomaly outside the North Pacific basin and nudging region, such as over north-eastern Siberia and south-central USA. Furthermore, while the extratropical SST anomalies are somewhat larger in NUDGED, particularly in the subpolar gyre, the tropical Pacific signal is substantially weaker by a factor of ~3."

Figure 2: Why would you expect this to be different depending on the year of the simulation? It doesn't seem essential to the overall paper to show this for three different averaging periods.

Because we are imposing a fixed, annually-repeating forcing, which is idealised, we wanted to explore the time dependence of the response to see whether the short-term response in years 1-2 and 3-4 differs from the long-term response. It was possible, for example, that we may 'build up' a spurious response in the ocean by imposing the same forcing year-on-year for 30 years. From the plots it is clear that the signal is already evident in years 1-2, and remains approximately consistent throughout the run with some local exceptions (e.g. JJA central Pacific years 3-4 vs. years 5-30).

Figure 4: There is much more that could be said about this figure. The DJF panel shows that heat fluxes from the atmosphere into the ocean are imprinting onto the SST pattern during the nudging period, while the positive heat fluxes in the KOE region in all other

seasons show that the cold SST anomalies in this region are reducing the heat loss to the atmosphere.

We agree and have added the following:

"The positive heat fluxes exhibited in the KOE region in all seasons outside of DJF are evidence that cold SST anomalies in this region reduce heat loss to the atmosphere throughout the simulations. Regions such as those in the north-east North Pacific appear to dampen the SST anomalies during MAM and JJA, which may indicate limited dynamic feedback to the atmosphere. However, across the central North Pacific, the persistence of surface latent flux anomalies year-round is expected given the surface temperature persistence and alludes to ocean-atmosphere feedbacks."

---

## Author Response (AR2)

*Dear Dr. Dow and co-authors:*

*Thank you for your thoughtful responses to the concerns and comments of the reviewers on your manuscript. Reviewer #2 has remaining concerns that must be addressed before the paper can be accepted for publication in WCD (posted to you, I believe). Below I add my concerns which should be addressed, and some suggestions to make text. Line numbers refer to the uploaded pdf named "egusphere-2023-1595-manuscript-version3.pdf"*

*Regards, David*

We thank the Editor for the further feedback and for collating additional comments from the reviewers. Below are, first, the responses to the Editor's comments, and second, the responses to the comments of reviewer 2. The responses below are in blue. We hope you find the revised manuscript improved and ready for publication in WCD.

*In his original review, Reviewer 2 asked: "What does it mean to express the anomalies between NUDGED and CONTROL "per standard deviation of the PDO index"? And how does one compare the amplitudes between the two?" Your response (reproduced next in blue) was very helpful and I strongly urge you to include this in the text – and to add similar text to the caption of Figure 1.*

**The anomaly between NUDGED and CONTROL is projected onto the first EOF from the control run to generate a pseudo-PC. The anomaly is divided by the pseudo-PC to calculate the anomaly per standard deviation of the PDO index expressed in a similar way to that derived from CONTROL.**

*As suggested this text has been included in the manuscript and the caption of Figure 1*

*The schematic in Fig. 7. is somewhat misleading: The negative SLP anomaly north of the equator is centered near ~30N and not 10N as presently shown. This is important because south of the maximum (near 10-20N) there will be anomalous westerlies – reduced trade winds, and thus reduced evaporation. Also, the red arrows just south of the equator that point northward are inconsistent with the SLP field (they should point southward). Finally, the text in the figure caption should be sharpened to avoid confusion (in particular, subtropics usually refers to ~10-30N, while extratropics includes the midlatitudes and subpolar regions). I suggest replacing the caption with the following (or something like it):*

*"**Figure 7:** Schematic depicting the mechanisms involved in the tropical SST anomalies manifest as a result of an intensification of the AL. An intensified AL (dashed black line) imposed during boreal winter is associated with westerly anomalies (reduced easterlies; solid red arrows) in the*

*subtropics and downward latent heat transfer. The migration of the SST anomalies southward during boreal winter is associated with westerly anomalies in the subtropics (reduced trades). The westerly anomalies act to weaken the background trades (filled red arrow) which reduces latent cooling due to decreased evaporation and hence an increase in subtropical Pacific SSTs. In the season after nudging, the temperature asymmetry about the equator induces an SLP gradient (solid black line, posiEve SLP; dashed black line, negaTIve SLP) that drives southerly winds across the equator. The Coriolis force acts to turn the southerly winds in the southern hemisphere westward and in the northern hemisphere eastward. When these anomalous winds are imposed on the background easterly trade winds (filled red arrows), the southerlies south of the equator increase the wind speed and therefore evaporative cooling, whilst north of the equator the background trades are weakened, reducing evaporative cooling. The westerly wind anomalies along the equator deepen the thermocline in the eastern tropical Pacific (red dotted line) and reduce upwelling/divergence of cooler waters at the equator."*

*Thanks for this feedback - the figure and the caption have been updated as suggested*

*The paragraph on lines 73-93 is not relevant and is a distraction for the reader. Please remove it.*

*Done*

*There is some sloppiness in Eqns. 1-6 that need to be fixed. I will attach at the end of this document a page that will help.*

*Thanks for sending this through - the equations have been amended as suggested*

*Line 154, change to read "within the nudging period (d = 0 is 15 Jan)".*

*Done*

*Line 158-162, this is confusing. How the amplitude of the imposed anomaly compares to the maximum amplitude in ERA5 isn't helpful. What is relevant is how the variability in the CONTROL compares to the variability in ERA5. With this in mind, I suggest you change the text on line 158 to read*

*"... with an NPI anomaly of-10.76 hPa, or -3.02 s, where s = 3.53 hPa is the standard deviation ..." , change the text on line 161-2 to read "... reanalysis data from 1979-2020, a 1 s NPI anomaly is 5.20 hPa.", and change the text on line 163 to read "... conducted using a comparably sized NPI anomaly in reanalysis data."*

*This point was also noted by Reviewer 2. The authors agree and have changed the text accordingly.*

*Line 223, "... Pacific Ocean ..."*

*Done*

*Line 275, change to read "... There are positive (downward) ..."*

*Done*

*Line 279-282, change this sentence to read "The pattern of surface latent heat flux anomalies in JJA in the extratropical North Pacific resembles the SST pattern associated with the internal PDO (Fig. S1d) and represents a damping of the SST anomalies; positive flux anomalies extend eastward from the KOE region, which are enveloped by negative anomalies in the northeast Pacific and subtropical North Pacific. The ... "*

*Done*

*Line 299, change to read "...zonal wind anomalies represent a ..."*

*Done*

*Line 331, change to read "...from the surface in the northern subtropics due to reduced..."*

*Done*

*Line 340-3421, the NPO is an intrinsic mode of atmosphere variability, not an intrinsic coupled atmosphere-ocean mode. Change to read "...the North Pacific Oscillation (NPO), but they imposed ..."*

*Done*

*Line 350, change to read "...coincides with an anomalous northward..."*

*Done*

*Line 358, replace "Investigation into" with "The"*

*Done*

*Line 355, change to read "…the warming in the central near-equatorial Pacific…"*

*Done*

*Figure caption 3: change "NUDGED-CONTROL" to read "NUDGED minus CONTROL". Also, add the text "The subtropical North Pacific and Nino3.4 domains are indicated by the boxes in Fig. 1".*

*Done*

*Figure caption 6: change "NUDGED-CONTROL" to read "NUDGED minus CONTROL".*

*Done*

*Figure caption S1: change "Seasonal mean surface" to read "Seasonal mean skin"*

*Done*

*Figure caption S4: What are the box limits? The whisker limits? Presumably 10, 25 75 and 90%, but best to state this explicitly rather than making the reader guess. The text on line 36 in parentheses is confusing. Suggest replacing this text with the sentence: "The maximum and minimum values of Nino3.4 in the HadISST4 and Control run are indicated by an "x" (and then put the x's on the plot.*

*Done*

Author Response to Reviewer 2 Further Comments

Thanks for engaging constructively with the review process and improving the framing of the manuscript. The title and abstract much more accurately reflect the results of the study and connect it to the state of the literature. While the model used is borderline inadequate for the questions investigated, this is at least caveated, so I don't think this needs to preclude publication. I have some more additional (mostly) minor comments to address before publication.

We thank the reviewer for the further constructive comments on the revised manuscript. We have taken these into account in the resubmitted version. We hope the reviewer finds the manuscript improved and ready for publication in WCD.

Major(ish) comments:

1. Your explanation of the quantification of statistical significance on lines 170-178 is quite sloppy and I think it is done incorrectly. You talk about multiplying by sqrt(2) to correct for the difference of 30-year means, but you should also divide by sqrt(2) to go from 15-year to 30-year averages. Technically this should also account for the autocorrelation of subsequent 15-year averages, but it would be easier just to use 30-year means of the control run instead. An even bigger omission is that you should also divide by sqrt(50) to account for the ensemble average. Slightly concerning that this would make almost everything significant, but 50 members is quite a large ensemble.

Thanks for raising this issue. Given the points raised in the comment, we have revised the approach for estimating statistical significance. Instead of comparing the ensemble mean response to unforced variability derived from CONTROL, we have instead computed the standard error on the ensemble mean using the NUDGED ensemble spread. This enables us to account for different averaging periods used in different parts of the analysis (e.g. Figure 2 shows different averaging periods from Figure 4) and overcomes the issues with estimating low frequency variability from the relatively short CONTROL run. We now estimate the significance of the ensemble mean anomaly by computing the standard error of the mean:

$$SE = \sigma/\sqrt{n}$$

where $\sigma$ is the inter-ensemble standard deviation of the time averaged quantity of interest (e.g. 30 year mean surface temperature) and n is the ensemble size, 50. We then identify where the ensemble mean NUDGED minus CONTROL anomaly ±2xSE is different from zero. This achieves a similar outcome to the point raised in the comment that the significance of the ensemble mean response is increased. The revised method is described in the Section 2.2.

2. Section 3.3: It is not clear from your figures that any of the wind anomalies are significant other than the ones in DJF and during the ramp up/down in November/March, unless you are only meaning to draw attention to the anomalies outside the nudging region in the subtropics.

Note that "persistent anomalies extending into the spring after nudging ceases (MAM)" is not true unless you remove March from the averaging, since nudging is still active in March. Perhaps you could instead show the anomalies from what would be expected if all you saw was the imposed nudging. When mentioning the upper tropospheric anomalies in Figure S5, it would be worth reminding readers that the nudging is smaller in the upper troposphere.

During the analysis, the results from separate months were analysed (not shown in the manuscript) and the anomalies referred to here do persist into April after nudging has ceased. We have amended the sentence to specify we mean April for clarity. We have also added a sentence referring to the weaker upper tropospheric nudging in the description of Figure S5: *"Recall that the nudging strength in the upper troposphere is several times weaker than at the surface (Fig. S2), so the upper-level circulation anomalies likely represent a response to the lower tropospheric forcing."*

3. Line 391-396: Nothing you say here is inconsistent with Klavans et al. (submitted, https://staff.cgd.ucar.edu/cdeser/docs/submitted.klavans.human_emissions_pdo.apr23.pdf), who also not the small amplitude in models and suggest some reasons for it. Please remove this reference from this final paragraph, because your study does nothing to refute it (Note that I am in no way affiliated with the Klavans study). The references to Klavans et al. elsewhere were okay.

Thanks for highlighting this - reference has been removed from this sentence.

Remaining minor comments:

• Line 73: I had to read this sentence several times to make sense of it, and I think the problem is that "via" should be replaced with "by" - Done

• Section 2.1: It might help to add a sentence of two in this section why this low-resolution model was used, though caveats added later are sufficient.

We have added *"While the model is run at relatively low horizontal and vertical resolution, the model code is sufficiently flexible to apply the nudging method described in Section 2,2 and the model is computationally efficient to run enabling a large ensemble to be produced."* to the end of Section 2.1.

• Eq. 5: spurious dot after a - removed

• Line 161: Based on the values in hPa, it is clear that the sigma is not the same here as defined on line 158. Please clarify this in the text. It would be cleaner (and more robust statistically) to state what the corresponding ~3 sigma value would be in ERA5, rather than stating the most extreme value.

This section has been corrected by changing the text to:

"Comparing with ERA5 reanalysis data from 1979-2020, a $1\sigma$ NPI anomaly is 5.20 hPa. The imposed atmospheric forcing is therefore weaker than if an equivalent experiment was conducted using a comparably sized NPI anomaly in reanalysis data."

• Line 168: Given the use of the long-term climatology of CONTROL here, it would be good to make a quick check to make sure there is not long-term drift in CONTROL, which could affect the results.

Thanks - this was checked at the outset of the study and the first 150 years of the run were discarded for spin-up/drift.

• What does LR mean in several of the figure titles? It is never defined.

Linear Regression - it is now defined in the caption for Figure 1 and S1.

• Figure S1: I think you mean HadSST4, which is used in HadISST1. Something looks very off with the patterns in HadSST4; have you removed the linear trend (if so, Section 2.4 doesn't say so)?

You are correct - these are HadSST4 monthly anomalies (not detrended) for the last 100 years. The $1^{st}$ EOF of SST anomalies in the PDO region is calculated and the SST anomalies are regressed onto the corresponding PC1.

• Just to check, for Figure 1, you are computing the EOF from SSTs and then showing the associated surface air temperature anomalies? If you computed the EOF from surface air temperatures (including land), then this is not okay and needs to be updated.

You are correct - the $1^{st}$ EOF from SSTs inside the region defining the PDO is computed then the surface temperature anomaly derived from regressing surface temperature anomalies onto PC1 is what is shown.

• Figure 2: I strongly caution against discussion SON as an initial response. From comparing Figure 2a and 2e, it is clear that almost the entire signal in (a) is coming from year 2, where the anomaly persists from the previous DJF.

Analysing the first year in isolation showed (not shown in manuscript) that there is a response in SON in year 1 (due to nudging starting in November) – therefore we are comfortable with the term 'initial response'.

• Line 250-255: You need to state explicitly when you compare with observations that the value of the tropical response in the nudged runs is ~10x weaker than the value in the observed PDO variability.

This has now been stated explicitly.

• Line 268-273: missing reference to Figure 3c,d

Added.

• Figure 3c,d: It is not clear what the shading is showing

The shading was a visual aid and has now been removed to avoid confusion.

• Line 279-280: Figure 4 does not show the surface heat flux anomalies associated with the internal PDO, which is what your text on this line indicates.

To avoid confusion we have removed the clause "resembles the SST pattern associated with the internal PDO" from the sentence.

• Line 285-286: Not clear what is meant by "limited dynamical feedback to the atmosphere"

• Figure 5: Please state what is meant by near-surface. 10 meter? Lowest model level?

Done.

• Figure S5: Which season?

DJF - Added in caption.

• Line 305: it could be helpful to remind readers that this is showing anomalies that are entirely outside of the nudging region

Thanks for the suggestion – we have added this to the text.

• Line 368: "Specifically" is repeated with the last sentence and doesn't really fit here. Suggest a use of also or furthermore instead, since this is a separate point from the ocean vertical resolution.

Have changed to "Furthermore"